# Enhancing Precision in Photodynamic Therapy: Innovations in Light-Driven and Bioorthogonal Activation

**DOI:** 10.3390/pharmaceutics16040479

**Published:** 2024-03-31

**Authors:** Natalia S. Kuzmina, Ekaterina A. Fedotova, Petar Jankovic, Galina P. Gribova, Alexander V. Nyuchev, Alexey Yu. Fedorov, Vasilii F. Otvagin

**Affiliations:** Department of Organic Chemistry, Lobachevsky State University of Nizhny Novgorod, Gagarina Av. 23, 603950 Nizhny Novgorod, Russia; kuzmina.ns2014@gmail.com (N.S.K.); necatirhe@gmail.com (E.A.F.); petar.jankovic96@mail.ru (P.J.); nastfilbar@gmail.com (G.P.G.); alex.nyuchev@ya.ru (A.V.N.)

**Keywords:** photosensitizers, photodynamic therapy, active targeting, bioorthogonal chemistry, sensing mechanisms, combination therapy, selective photodynamic therapy

## Abstract

Over the past few decades, photodynamic therapy (PDT) has evolved as a minimally invasive treatment modality offering precise control over cancer and various other diseases. To address inherent challenges associated with PDT, researchers have been exploring two promising avenues: the development of intelligent photosensitizers activated through light-induced energy transfers, charges, or electron transfers, and the disruption of photosensitive bonds. Moreover, there is a growing emphasis on the bioorthogonal delivery or activation of photosensitizers within tumors, enabling targeted deployment and activation of these intelligent photosensitive systems in specific tissues, thus achieving highly precise PDT. This concise review highlights advancements made over the last decade in the realm of light-activated or bioorthogonal photosensitizers, comparing their efficacy and shaping future directions in the advancement of photodynamic therapy.

## 1. Introduction

The therapeutic influence of light on living organisms has been known since ancient times. However, the development of PDT began in the early 20th century with the studies of H. von Tappeiner and O. Raab, who explored the light impact on infusoria incubated in acridine red. This marked the emergence of PDT, based on the combination of light, a photosensitizer (PS), and oxygen [1,2]. In the second half of the 20th century, the research by S. Schwartz, R. Lipson, and T. Dougherty, focusing on the oligomeric mixture of hematoporphyrin derivatives (HpD), contributed to the advancement of PDT as a method for anticancer therapy. They investigated HpD as a diagnostic and therapeutic agent, localizing in tumor cells and inducing their fluorescence [3,4,5].

Modern photodynamic therapy is a minimally invasive treatment for various dermatological, ophthalmological, dental, cardiovascular, gynecological, infectious, and oncological diseases. The classical PDT procedure involves two stages: intravenous or topical PS administration, distribution throughout the body, accumulation in tumor tissues, and then activation of PS by light of the corresponding wavelength (Figure 1). A time gap, typically spanning several hours, exists between the administration of the PS and the exposure to light, referred to as the drug–light interval (DLI) [6,7].

Due to its relatively high effectiveness, PDT can be applied as a standalone therapy or in combination with other treatment methods [8]. Among its main advantages are patient-friendly treatment (low dark toxicity of drugs, minimal side effects, and rapid tissue recovery) and the potential to stimulate an antitumor immune response. Recently, the elicited anticancer response induced by PDT has drawn significant attention, as it leads to immunogenic cell death (ICD) [9]. During cell death induced by PDT, intracellular components known as damage-associated molecular patterns (DAMPs) are released. After recognition of DAMPs by pattern recognition receptors (PRRs) expressed on immune cells, activation of the T-cell adaptive immune response and long-term immunological memory occur [10].

Current limitations of PDT include challenges in treating hypoxic tumors, deeply located, massive, and metastatic lesions, limited light penetration, post-procedure photosensitivity of the skin, and the need for individualized treatment protocols for each patient [6,7,11].

PDT is based on the interaction of three separately non-toxic components: a photosensitizer, light of a specific wavelength, and molecular oxygen (Figure 2A). The Jablonski diagram (Figure 2B) is used to describe the physicochemical processes occurring among them. When exposed to light, the PS molecule in the ground state (S_0_) absorbs a photon, transitioning to the excited singlet state (S_1_). In this state, the PS exists for a few nanoseconds [12], after which excess energy is dissipated through fluorescence (radiative relaxation) or internal conversion with the release of heat (non-radiative relaxation) [6,13]. Additionally, for PS molecules in the S_1_ state, intersystem crossing (ISC) to the excited triplet state (T_1_) is possible. T_1_ has a relatively long lifetime (up to several tens of microseconds), associated with a prolonged spin inversion to return to the S_0_ state [6,12]. The relaxation from T_1_ to S_0_ occurs either through phosphorescence (radiative relaxation) or via energy transfer to molecules in the surroundings, such as molecular oxygen. Energy/electron transfer processes lead to the formation of singlet oxygen (^1^O_2_) and other reactive oxygen species (ROS), which are highly potent oxidants [14]. Subsequently, ROS species (^1^O_2_, superoxide ion, hydrogen peroxide, and hydroxyl radical) interact with biomolecules (proteins, DNA, RNA, and lipids), causing oxidative stress [14,15,16].

It is noteworthy that the organism lacks natural defense mechanisms against singlet oxygen, whereas superoxide anion radical and hydrogen peroxide are rapidly eliminated by superoxide dismutase and peroxidase, and hydroxyl radicals can be captured by intracellular radical traps (glutathione (GSH), vitamins A, C, and E) [17,18].

Various forms of cell death are induced by photodamage, including apoptosis, necrosis, and autophagy. Recent studies have identified additional modes of cell death triggered by PDT, such as regulated necrosis variants like necroptosis, ferroptosis, pyroptosis, parthanatos, and mitotic catastrophe (Figure 3) [9,15,19]. The specific type of cell death depends on factors such as the localization of the photosensitizer within organelles, the concentration of the photoactive agent, the DLI, and the irradiation dose [13]. Moreover, photodynamic reactions may activate multiple types of cell death concurrently, significantly influencing the therapeutic effectiveness of PDT.

### Photosensitizers

As mentioned earlier, photosensitizers play a crucial role in the mechanism of PDT. Their most important property is the ability to absorb light of a specific wavelength, thereby initiating photochemical reactions with molecular oxygen [8]. However, not every natural or synthetic PS is suitable for clinical use. Hence, in the quest for developing new photoactive drugs, researchers endeavor to synthesize compounds possessing the characteristics of an ideal photosensitizer (Figure 4).

In the development process, the primary goal is to predict the photochemical and photophysical properties of the prospective PS. It should possess an absorption peak (maximum) within the range of 600–800 nm, known as the “phototherapeutic window”. Additionally, strong fluorescence is desired for potential utilization as an imaging agent in tumor diagnostics, along with a high quantum yield of singlet oxygen or other ROS. To ensure patient safety, factors such as the speed and selectivity of drug accumulation, potential toxicity in the absence of irradiation, and immediate post-procedure effects must be carefully considered. Conversely, the production of such a PS should be rapid, cost-effective, and yield a chemically pure, stable, and commercially available product [8,19,20].

At present, there is no photosensitizer in clinical practice that meets all these characteristics. Nevertheless, there are several photosensitizing drugs that have demonstrated success in the therapy of oncological diseases [20].

The most common class of photosensitizers is tetrapyrrole compounds or porphyrins, which possess unique photophysical properties due to their extensive conjugated π-system [20,21]. Certain classes of non-porphyrin photosensitizers, such as boron complexes of dipyrromethenes (e.g., 4,4-difluoro-4-bora-3a,4a-diaza-s-indacene, abbreviated as BODIPY) and their aza-analogues, cyanines, anthraquinones, phenothiazines, and xanthene dyes, have also gained significant attention. This is mainly due to their ease of structural modification, enabling adjustment of the light absorption maximum towards the near-infrared (NIR) region [22]. Based on differences in chemical structure and characteristics, photosensitizers are classified into three generations (Figure 5).

First-generation PSs 

The first generation of photosensitizers comprises HpD, historically the initial porphyrin-based photosensitizers studied on tumor cells and subsequently employed in human applications. HpD is an oligomeric mixture obtained through the hydrolysis of hematoporphyrin, isolated from blood hemoglobin with concentrated sulfuric acid (H. Scherer, 1841) [12]. The photochemical activity of HpD is attributed to its oligomeric forms containing 2–9 porphyrin rings linked by ether and C-C bonds [23]. HpD, however, exhibits several drawbacks including unsatisfactory photophysical characteristics, rapid aggregation, and low quantum yield of singlet oxygen, as well as toxicity and a prolonged accumulation period.

Purified from inactive monomer forms, Photofrin, developed by T. Dougherty, is considered the gold standard in PDT. Despite its successful therapeutic application, Photofrin necessitates high-dose administration due to its low molar extinction coefficient. Additionally, Photofrin has a tendency to aggregate in the aqueous phase, leading to reduced selective accumulation and prolonged skin photosensitivity [8,20,24,25].

Second-generation PSs 

To address some of the limitations of first-generation photosensitizers, new porphyrin-like photosensitizers have been developed, including benzoporphyrins, chlorins, purpurins, phthalocyanines, and others. Commercial photoactive agents belonging to the second generation are listed in Table 1. These compounds represent chemically purer and more homogeneous substances with improved photophysical characteristics, such as higher singlet oxygen yield and absorption maxima in the range of 650–800 nm. They demonstrate minimal phototoxicity to tissues and fewer overall side effects compared to their predecessors. Moreover, second-generation photosensitizers exhibit greater selectivity for accumulation in tumor masses and faster elimination from the body [8,19,24,25].

Among second-generation photosensitizers, chlorins and phthalocyanines stand out due to their highest extinction coefficients at absorption maxima within the phototherapeutic window (650–900 nm). This spectral range aligns with maximum light transparency in tissues, minimizing damage to vital cellular components. Water-soluble derivatives can be easily derived from naturally occurring chlorin PSs, while phthalocyanine PSs exhibit photostability and straightforward chemical modification [26,27].

However, a significant limitation of second-generation PSs is their limited selectivity for accumulating in tumor cells. Additionally, these PSs have limited solubility in water, leading to aggregation under physiological conditions and suppressing the generation of ROS. This constraint restricts their intravenous administration, necessitating the exploration of novel approaches for drug delivery to tumor sites [8,24].

Third generation 

Third-generation photosensitizers were developed as targeted drug-delivery agents directly to tumor tissues, bypassing the possibility of accumulation in healthy ones. This is achieved by conjugating or encapsulating second-generation photosensitizers with compounds that have an affinity for tumor cells or their expressed receptors [8,19,28]. Biomarkers such as peptides [29], carbohydrates [30], vitamins [31], growth factor receptors [32], antibodies [33], and others serve this purpose. This approach enhances the pharmacokinetics, pharmacodynamics, and bioavailability of the photosensitive drug, improves water solubility, and stabilizes the conjugate during use [24]. However, the application of such PSs is still limited and current research is actively seeking new options for controlled drug delivery to tumor-affected tissues.

Our group is also involved in creating multifunctional photosensitizing systems. For instance, water-soluble conjugates of chlorin-*e_6_* derivative and arylaminoquinazoline, an inhibitor of EGFR and VEGFR (epidermal and vascular endothelial growth factor receptors, respectively), have been developed [34,35,36]. We also present a cleavable conjugate sensitive to β-glucuronidase, based on a zinc complex of a chlorin-*e_6_* derivative and the cytostatic cabozantinib, which inhibits the tyrosine kinases VEGFR-2 and c-Met [37]. Among the new methods of selective drug delivery, the concept of linking drugs with light-sensitive bonds is intriguing, as such an external stimulus allows for precise spatiotemporal control over drug release. Recently, in our group, a model conjugate of a porphyrin-based PS and the antimitotic agent *trans*-combretastatin A4, connected by a photocleavable *o*-nitrobenzyl linker, was synthesized [38].

In recent decades, significant strides have been made in advancing the notion of activating photosensitizers directly within specific target tissues. To achieve this, the photosensitizer is switched into an “off” state of its photoactivity, primarily through various mechanisms of electron/charge transfer with an additionally introduced fragment of the photosensitizer [39]. This introduced fragment can also serve other functions, such as acting as a chemotherapeutic agent, enabling the synergistic PDT and chemotherapy. Activation of the “turned-off” photosensitizer is typically achieved with tumor-specific stimuli, allowing for a more selective PDT. An innovative approach to enhance the selectivity of photosensitizer accumulation in tumor cells has recently been implemented using bioorthogonal delivery, facilitating rapid binding to the surface of tumor cells [40].

This review discusses three currently relevant trends in photosensitizer development: the combination of PDT and controlled light drug delivery, the creation of activatable photosensitizers, and the selective delivery of photosensitizers through bioorthogonal chemistry. These directions aim to enhance the selectivity of PDT and create third-generation drugs.

## 2. ROS-Activated PS–Drug Conjugates

### 2.1. Aminoacrylate Linker

The reactive oxygen species produced upon PS irradiation can serve dual roles, not only as cytotoxic agents interacting with cellular components but also by interacting with sensitive bonds within a hybrid conjugate of the photosensitizer and the drug. This interaction may result in the liberation of the active form of the drug. In this case, a synergistic approach combining PDT and chemotherapy is implemented for the treatment of neoplastic diseases.

Significant breakthroughs have been achieved in the development of a β-aminoacrylate group sensitive to singlet oxygen, led by Y. You’s research group (Figure 6) [41]. This system operates through a well-known [2+2]-cycloaddition reaction involving singlet oxygen produced by the PS during irradiation and an alkene. Generally, the interaction of singlet oxygen and an alkene produces the dioxetane intermediate, which undergoes decomposition, resulting in the production of two carbonyl compounds (Figure 6a) [42].

It was suggested to employ the described system for the controlled release of the hydroxyl-containing active drug by attaching the hydroxyl group to the C-C double bond through an ester linkage. In this approach, the photosensitizer was linked to the C-C double bond via an amino group (Figure 6b). The resulting β-aminoacrylate linker demonstrated stability in the cellular environment without irradiation, swiftly reacted with singlet oxygen, and efficiently released chemotherapeutic agents (SN-38, combretastatin A-4 (CA4), paclitaxel (PTX)) [43,44,45]. For example, the *cis*-combretastatin A-4, when linked with a β-aminoacrylate linker to the dithioporphyrin PS with the formation of conjugate **1**, did not exhibit inhibitory activity against tubulin polymerization compared to free combretastatin A-4 **2** (6% inhibition compared to 100% inhibition of CA4) [44] (Figure 7A). Meanwhile, the cytotoxicity of the cleavable conjugate **1** increased by six orders of magnitude under light exposure (from the half-inhibitory concentration IC_50 dark_ = 164 nM to IC_50 light_ = 28 nM, MCF-7 cells) compared to a 1.7-fold increase in the activity of a structurally similar non-cleavable conjugate (IC_50 dark_ = 1802 nM to IC_50 light_ = 1063 nM). The obtained results indicated the expression of a chemotherapeutic effect of free combretastatin A-4. Also, the bystander effect was observed in cells incubated with **1**, which manifested as the presence of affected cells around the irradiated area. This indicates that cellular damage resulted from the released CA4 rather than singlet oxygen.

In 2014, Y. You et al. presented a study in which a porphyrin PS was replaced with a silicon phthalocyanine, capable of strong absorption in the red region of the visible spectrum and possessing a high singlet oxygen quantum yield [46]. A remarkable antitumor effect was observed in vivo: within 15 days, the tumor almost completely disappeared in the mouse group (Balb/c mice with SC tumors) subjected to light treatment with the photocleavable conjugate. Meanwhile, in the group receiving the non-cleavable conjugate, tumor reduction occurred only within 3 days of light treatment, after which the tumors returned to control group levels. The next step in applying the combined system based on the β-aminoacrylate linker was the inclusion of a folate acid-targeting moiety in the phthalocyanine–combretastatin conjugate. Folic acid was bound to the PS using PEG linkers of various lengths to increase the conjugate’s hydrophilicity [47]. Thus, the cleavable conjugate with the longest PEG fragment (approximately 45 units) showed the highest cellular accumulation and, consequently, the highest phototoxicity (IC_50 light_ = 16.5 nM, colon 26 cells, λ_irr_ = 690 nm at 5.6 mW/cm^2^ for 30 min (10 J/cm^2^)). The conjugate also demonstrated selective accumulation, induced by binding to folate receptors, while the comparison conjugate without folate acid exhibited less specific binding to tumors. Due to its selective accumulation in tumors in mice with colon tumors the conjugate induced a strong antitumor effect (mice were cured of tumors by day 75) with minimal damage to healthy tissues.

In 2020, D. K. P. Ng et al. aimed to enhance the precision and selectivity of the phthalocyanine–combretastatin–aminoacrylate linker system by incorporating a fluorescence quencher through a glutathione-sensitive linkage (Figure 7) [48]. Conjugate **3a** is composed of zinc phthalocyanine linked with an β-aminoacrylate linker to *cis*-combretastatin A-4 and a target fragment based on biotin. The 2,4-dinitrobenzenesulfonate (DNBS) quencher in **3a** prevented singlet oxygen and fluorescence generation. However, upon removal of the quencher by glutathione, the PS regained its activity. Under light exposure (300 W halogen lamp, λ = 610 nm) of HepG2 cells (positive for the biotin receptor), the cytotoxicity of the conjugate **3a** significantly increased (from IC_50 dark_ = 2 μM to IC_50 light_ = 48 nM), whereas the conjugate **3b** without the aminoacrylate linker exhibited reduced toxicity under both dark and light conditions. Additionally, the authors calculated a combination index (CI) [49], using the dose-dependent survival curves of free CA4, **3a**, and **3b** against HepG2 cells. In the range of IC_20_ to IC_75_, a synergistic action of the conjugate **3a** (CI < 1) was observed.

The thio-4-dimethylaminonaphthalimide photosensitizer, known for its high quantum yield of singlet oxygen (Φ_Δ_ > 85%), was conjugated with the aforementioned drugs (SN-38, combretastatin A-4, and paclitaxel) using a β-aminoacrylate linker to form combined conjugates [50]. 

All conjugates exhibited toxicity at low nanomolar concentrations upon irradiation. The capability of the β-aminoacrylate linker to undergo disruption upon interaction with singlet oxygen was exploited for the concurrent release of the encapsulated photosensitizer and chemotherapeutic agent from “smart” nanoparticles formed by an amphiphilic PEG copolymer. Singlet oxygen, generated during PS irradiation, facilitated the disruption of the nanoparticles, leading to the separate release of the PS and the therapeutic agent. As a result, the photoactivity and fluorescence of the photosensitizer increased. Furthermore, a bystander effect (killing neighboring cells) was observed under light irradiation in HepG2 and MCF-7 cancer cells incubated with conjugates [51,52].

### 2.2. Thioketal Linkage

Among the photoactivatable linkers activated by reactive oxygen species, thioketal (thioacetal) has gained widespread use in recent decades due to its easy synthesis, rapid metabolism, and non-toxic byproducts. It has found applications in the treatment of cancer and inflammatory diseases [53]. Various ROS (superoxide anion radical, hydroxyl radical, and hydrogen peroxide) react with the thioketal moiety, resulting in the release of corresponding non-toxic thiols, acetone, and oxygen (Figure 8) [54]. The advantages of the thioketal linker include its stability under acidic and basic conditions, and resistance to enzymatic activity.

Using the thioketal linker, PS–drug conjugate **4** was created (Figure 9) [55]. It consisted of a BODIPY photosensitizer, absorbing light in the red region of the visible spectrum, and a topoisomerase I inhibitor, camptothecin (CPT) **5.** To enhance the selectivity of the PS’s action, a glutathione-sensitive DNBS quencher was introduced into **4**, forming a photoinduced electron transfer system. Thus, the fluorescence and singlet oxygen generation of **4** were quenched until the photosensitizer entered tumor cells, where excess glutathione facilitated quencher removal, and the BODIPY **6** regained its photoactivity.

Prodrug **4** penetrated cells aided by a biotin ligand that attached to biotin receptors present on pathogenic cells. Irradiation of the conjugate led not only to the precise activation of the photosensitizer **6** but also to the cleavage of the thioketal linker, resulting in the simultaneous release of the chemotherapeutic agent camptothecin **5**. To enhance the biocompatibility and hydrophilicity of the conjugate **4,** two triethylene glycol fragments were also introduced. In contrast to other systems with thioketal linkers, where the photosensitizer was in the activated form (turned on) [56,57,58], this strategy achieved a reduction in photodamage to surrounding healthy cells. Additionally, the combination of tumor-specific accumulation with fluorescence activation allowed for precise real-time monitoring of the therapeutic agent release. In vitro experiments on HepG2 and HeLa cells (λ = 660 nm, 20 mW/cm^2^, 6 J/cm^2^) showed that conjugate **4** was significantly more toxic under light conditions (IC_50 light_ = 0.29 µM in HepG2 cells and 0.21 µM in HeLa cells) than the conjugate lacking the biotin targeting fragment and the conjugates without the thioketal linker. The high phototoxicity of conjugate **4** was attributed to its preferential accumulation in the tumor due to the biotin fragment, as demonstrated by fluorescent molecular tomography (FMT). The authors attributed the high antitumor activity to the presence of the thioketal linker, which facilitated the camptothecin **5** release upon irradiation. Furthermore, compound **4** did not cause significant systemic toxicity in vivo under laser irradiation.

In 2023, L. Qiu et al. put forward an intriguing fusion of photodynamic therapy and chemotherapy, utilizing an internal radiation source for the activation of the photosensitizer through Cherenkov radiation [59]. The use of Cherenkov radiation (λ = 300–500 nm) instead of the traditionally employed external light source, such as gamma-ray emission from gallium-68 (^68^Ga), addresses a critical challenge in PDT—the limited light penetration in living tissues, complicating the treatment of solid tumors. To validate their concept, the authors created conjugate **7**, consisting of tetraphenylporphyrin linked to the chemotherapeutic agent camptothecin **5** using a thioketal linker (Figure 10). After internalization, the conjugate **7** was activated by the radiation from the radionuclide, resulting in the generation of reactive oxygen species that destroy the thioketal linkage, leading to the release of free chemotherapeutic drug **5**.

To check the cytotoxicity of conjugate **7**, it was essential to selectively deliver the Cherenkov radiation source to the tumor cells. For this purpose, the radiotracer [^68^Ga]Ga-NOTA-Nb109 was chosen. Due to the presence of the single-domain antibody Nb109, linked to radionuclide ^68^Ga through a NOTA chelator, [^68^Ga]Ga-NOTA-Nb109 could selectively accumulate in human melanoma A375-hPD-L1 cells with overexpression of programmed death-ligand 1 (PD-L1) [60]. Conjugate **7** was almost non-toxic in the absence of the radiotracer and in the dark. However, in the presence of [^68^Ga]Ga-NOTANb109 (dose of 11.1 MBq), half-inhibitory concentrations were 0.5 μM for compound **7**. At this concentration, only 33% of cells died from the reference conjugate lacking the thioketal linker, demonstrating the effectiveness of the combination of chemo- and photodynamic therapy. It is noteworthy that the efficacy of conjugate **7** was evaluated under both Cherenkov radiation and light irradiation. It was shown that, under light irradiation, the activity of the cleavable conjugate **7** was slightly lower than when activated by Cherenkov radiation.

J. Gao et al. developed a photosensitizer based on a cyanine dye with a triphenylphosphonium moiety, capable of binding to mitochondria (Figure 11, compound **9**) [58]. Performing vital functions in the cell, mitochondria are considered one of the primary targets in anticancer therapy [61]. As a therapeutic fragment, the authors chose camptothecin **5**, as it can inhibit DNA topoisomerase I during cancer treatment. Additionally, compound **5** acted as an inhibitor of cellular respiration, causing significant damage to mitochondria. The linkage between camptothecin and the photosensitizer was achieved using the thioketal linker. In vitro studies (IC_50 light_ = 1.6 μM under λ = 660 nm irradiation, 0.1 W/cm^2^ for 10 min in A549 cells) and in vivo investigations (nude mice bearing A549 tumors) confirmed the outstanding antitumor effectiveness of compound **9,** as well as its selective binding to mitochondria and initiation of apoptosis due to their damage.

A good alternative to covalent conjugates is nanosystems that allow the selective delivery of a photosensitizer and chemotherapeutic agent to tumor cells. In recent years, a series of nanoparticles have been created, consisting of a therapeutic agent and a photosensitizer linked by a thioketal linker, which have demonstrated their effectiveness by enhancing therapeutic efficiency and mitigating potential side effects [62,63,64,65].

## 3. Photoactivated Chemotherapy (Photocleavable Groups)

Like PDT, photoactivatable chemotherapy (PACT) is aimed at effectively destroying cancer tissues with minimal damage to surrounding healthy tissues. However, its path to achieving a therapeutic effect differs from PDT: in PACT, the cytotoxic inhibitor is masked by a photocleavable protective group (PPG) containing a labile chemical bond, which is broken by light, thereby releasing the cytotoxic agent (Figure 12) [66]. A significant advantage of this drug-delivery strategy is the independence of the photochemical reaction from the presence of molecular oxygen, which is particularly relevant in the treatment of hypoxic tumors, where clinically approved PDT agents are often ineffective [67].

The majority of known photosensitive groups, such as nitrobenzyl (λ_abs_ = 340–365 nm), coumarin (λ_abs_ = 310–490 nm), phenacyl (λ_abs_ = 270–340 nm), and BODIPY (λ_abs_ = 510–550 nm), are activated by UV or short-wavelength visible light, which can be harmful to healthy tissues and penetrate several millimeters deep [68]. Therefore, in recent years, efforts have been made to shift the absorption maximum of photosensitive groups into the red region of the visible spectrum and the NIR range (Table 2) [69].

Thus, PPGs based on ruthenium complexes release the therapeutic ligand through metal–ligand charge transfer (MLCT) [74]. Additionally, in this case, the ligand protects cells from the cytotoxic effect of the ruthenium atom. Photo-induced disruption of the metal–ligand bond leads to a toxic effect from both the ligand and ruthenium [75]. Until recently, ruthenium PPGs were active in both UV and the short-wavelength visible range (λ_abs_ = 350–480 nm) [76]. However, modifying the ligand design to extend the conjugated π-system and adding sterically bulky groups allowed shifting the photoactivation to the red region of the visible spectrum (λ_abs_ = 600–650 nm).

In the work of W. Sun et al., the primary metabolite of curcumin, tetrahydrocurcumin, known for its anticancer activity, was used as the ligand (Figure 13, compound **11**). Due to the extended conjugated π-system, tetrahydrocurcumin is well-suited as a ligand for the long-wavelength shift in absorption of the ruthenium PPG [77]. The authors proposed linking the ruthenium PPG with tetrahydrocurcumin using a photosensitive benzonitrile group in amphiphilic polymeric nanoparticles **11**, demonstrating NIR-triggered disintegration and activation of combined photo- and chemotherapy. However, the evaluation of conjugate **11** activity on MCF-7 and 4T1 tumor cell lines showed almost identical toxicity levels in the dark and upon irradiation (approximately 100 mg/mL at 760 nm, 0.2 W/cm^2^, 10 min, 120 J/cm^2^). Nevertheless, in vivo studies on 4T1 tumor-bearing mice with **11** showed a reduction in tumor volume and minimal systemic toxicity 14 days after irradiation (0.3 W/cm^2^, 10 min, 180 J/cm^2^), compared to the control group and the group receiving conjugate **11** without irradiation. The authors also demonstrated that the released ruthenium complex **12** could inhibit the mitogen-activated protein kinase signaling pathway characteristic of tetrahydrocurcumin, thereby exerting an anticancer effect.

In 2021, N. Toupin et al. proposed to link a ruthenium PPG with the pyridyl-BODIPY photosensitizer, which has a high quantum yield of singlet oxygen (Figure 14) [78]. Upon irradiation with green light (λ = 520–530 nm) for 15 min at an energy density of 50 J/cm^2^, the pyridyl-BODIPY ligand **15** was liberated from the conjugate **13**. The presence of heavy atoms in the BODIPY facilitated ISC, resulting in an increased quantum efficiency of the ruthenium PPG photorelease reaction, as previously noted for BODIPY PPG [79]. Irradiation of **13** led to quenching of fluorescence, and the half-inhibitory concentrations was of low micromolar values (IC_50 light_= 0.35  ±  0.04 μM) for triple-negative MDA-MB-231 cells, with an EC_50 dark_/EC_50 light_ ratio exceeding 100. Although the BODIPY **15** itself exhibited similar light toxicity, the conjugate with the ruthenium complex **13** 2.5 times more selectively bound to tumor cells compared to normal cells than BODIPY **15**.

The mentioned BODIPY photosensitizer itself can serve as a photocleavable group for the drug release if it has an O-containing leaving group in the meso-methyl position [80]. The work of A. Winter and R. Weinststein identified structure–property relationships for meso-methyl-BODIPY, allowing the absorption maximum shift into the red region of the visible spectrum, increased quantum yield of the photorelease reaction due to boron methylation, heavy atom effect, and the presence of methyl groups in positions 1,7 and 3,5 [81,82]. K. Zlatić et al. also demonstrated how changes in structural parameters affect the photoactivity of meso-methyl-BODIPY as a photosensitizer [83].

In 2019, N. Toupin et al. demonstrated an instance of photorelease of a therapeutic molecule from the BODIPY conjugate 16 (Figure 15) [71]. Green-light-triggered release of the cathepsin B (CTSB) inhibitor **18** led to a shift in the type of cell death in MDA-MB-231 triple-negative breast cancer cells from apoptosis to necrosis. Necrosis lacks adaptation mechanisms in cells and triggers the body’s immune response. The combined effect was manifested in the cytotoxic action of the conjugate **16**, also attributed to the singlet oxygen generation from BODIPY **17**. The phototherapeutic index (light toxicity to dark toxicity ratio) of conjugate **16** was greater than 40 (t = 15 min, λ_irr_ = 460−470 nm), making the conjugate **16** a promising agent for combined therapy. This is attributed to preferential accumulation of **17** in tumor cells compared to normal cells.

However, a notable drawback of the employed BODIPY PPGs is their activation with green light of the visible spectrum, which has low cellular penetration. Winter et al. introduced a range of red-absorbing BODIPY derivatives. However, they are less efficient as photolabile groups and exhibit significantly lower quantum yields of the photorelease reaction (Φ_rel_ ≤ 10^–3^, compound **20**), compared to shorter-wavelength derivatives (Φ_rel_~0.1, compound **19**) (Figure 16) [81]. These compounds were synthesized by introducing styryl groups at the 1 and 7 positions of BODIPY and had excited states with lower energy and shorter lifetimes than their short-wavelength counterparts (for example, **19**). To enhance the release quantum yields of long-wavelength BODIPY derivatives, blocking nonproductive photodecay channels (B-F bond cleavage, photoisomerization, and charge transfer) to prevent access to nonproductive conical intersections was proposed [84,85]. For this purpose, more rigid derivatives **21** with a condensed ring structure were obtained, where there was no free rotation around the C-C bonds as in **20.** Derivatives **21** exhibited much higher quantum yields of photorelease (λ = 681 nm, Φ_rel_ = 3.8%) than distyrylBODIPY **20** [86]. However, despite red and near-infrared absorption, and high Φ_rel_ values for distyrylBODIPY and BODIPY with condensed rings, the in vivo investigation of drug photorelease from these PPGs, as well as their photodynamic properties, has not been explored yet.

A completely new approach to activate the photorelease of *meso*-methyl-BODIPY with longer-wavelength light was proposed by W. Wang et al. [87]. Their idea involved using a red-absorbing photosensitizer for triplet–triplet energy transfer (TTET) to a green-absorbing BODIPY PPG, followed by the photorelease reaction. For efficient TTET from one molecule to another upon irradiation, their triplet excited states must be close in energy and long-lived. Briefly, upon absorbing light in the red range of the visible spectrum, the photosensitizer goes from the ground state to the singlet excited state, followed by intersystem crossing to the T_1_ state of the PS. Subsequently, energy is transferred from this state to the T_1_ state of the prodrug, which undergoes photolysis (Figure 17A). Effective TTET requires anaerobic conditions, as oxygen can quench the excited triplet state, reducing the quantum yield of photolysis.

As a model system for photolysis, a platinum complex of tetraphenyltetrabenzoporphyrin **22**, with an absorption maximum at 635 nm, and a prodrug based on chlorambucil and *meso*-methyl-BODIPY **23** were selected. Prodrugs **22** and **23** were encapsulated in polymeric micelles to increase hydrophilicity and bioavailability (Figure 18). Moreover, the location of **22** and **23** inside the micelle, protected by a hydrophilic shell, reduces oxygen access to the micelle’s internal content. Therefore, TTET inside the micelle can be implemented under normoxic conditions in tumors. When HeLa cells were irradiated with λ = 635 nm light (60 mW/cm^2^, 10 min), micelles containing both the PS and the prodrug demonstrated high cytotoxicity with low dark toxicity. Simultaneously, micelles containing only prodrug **23** or only platinum porphyrin **22** were significantly less toxic under red light exposure. Concerning the photodynamic effect of micelles with the prodrug and porphyrin, the quantum yields of ROS generation were slightly higher than for micelles containing **22** and **23** separately. However, the ROS quantum yield values for all investigated groups were low, indicating the secondary role of photodynamic therapy in the observed cytotoxicity of micelles with the prodrug and porphyrin. In HeLa tumor-bearing nude mice, micelles with the prodrug and porphyrin demonstrated a significant reduction in tumor volume after λ = 635 nm light irradiation, with no significant side effects on the 11th day of treatment.

The application of the energy transfer concept between triplet states of the photosensitizer and the prodrug has been limited to the use of specific photosensitizers with a low singlet state energy level (e.g., palladium or platinum complexes of porphyrins), as the photon energy must be no lower than the S_1_ state of the PS. A solution to this problem was the use of PSs with singlet–triplet (ST) absorption (designated as STPS) (Figure 17B) [88]. Such photosensitizers can be directly activated to the T_1_ state from the S_0_ state, bypassing the S_1_ state. As the number of energy transfer steps decreases, internal energy losses also decrease, thereby increasing the efficiency of the photorelease process. Although direct population of the T_1_ state from S_0_ is spin-forbidden, it is achieved for some metal complexes due to strong spin–orbit coupling [89]. To initiate photolysis, a photon with energy above that of T_1_ STPS is sufficient, requiring lower excitation energy and achieving activation in the NIR-light range.

In the presence of osmium(II) complex **24** with bromophenylterpyridyl ligands as singlet–triplet absorption molecules, a series of short-wavelength photosensitive prodrugs conjugated with *meso*-methyl-BODIPY were tested for photorelease. This series includes the prodrugs chlorambucil **23**, vadimezan **26**, indomethacin **27**, naproxen **28**, ibuprofen **29**, benzyl oxycoumaric acid **30**, tetracaine **31,** dopamine **32**, tyramine **33**, and homoveratrylamine **34** (Figure 18). The mentioned prodrugs were activated with λ = 690 nm light (100 mW/cm^2^, 5 min) with high photorelease yields (up to η_rel_ = 87%). However, it is worth noting that a structurally similar prodrug to chlorambucil **25**, lacking iodine atoms at the second and sixth positions of BODIPY, did not undergo photolysis at all when using osmium complex **24**. This was due to inefficient population and rapid deactivation of the triplet excited state T_1_. The authors achieved an impressive result, as the efficiency of drug release through energy transfer from the photosensitizer to the prodrug was even higher (84% with **24** and 42% with **22**) than when exposing compound **23** to short-wavelength light (32%, λ = 530 nm). Thus, a strategy is proposed to use low-energy photons of long-wavelength light (λ = 635 or 690 nm) to initiate the photolysis of prodrugs with high yields, demonstrating the potential of enhanced photoactivated therapy.

Cyanine and xanthene groups stand out among various NIR-PPGs (Table 2). Cyanine chromophores, with high molar extinction coefficients and excellent biocompatibility, exhibit absorption within the phototherapeutic window [90]. The photorelease reaction of cyanine PPGs **35** depends on the presence of oxygen, in which the polyene chain undergoes cleavage (Figure 19) [91]. Subsequent hydrolysis leads to the breakdown of the C4-N bond in **36a,b**, releasing the leaving group.

In 2021, Zh. Guo et al. created a cyanine-based PPG designed for use in tumor hypoxia conditions [92]. The photorelease (λ = 670 nm) of the chemotherapeutic agent camptothecin from cyanine-based PPG was accompanied by the fragmentation of the cyanine platform and was demonstrated in mice bearing A549 tumors.

Xanthene derivatives are well known as dyes and have recently gained attention as photocleavable groups. Their advantages include synthetic accessibility, low molecular weight, and ease of modification of photophysical properties. For instance, in a recent study by M. Bojtár et al., water-soluble and red-absorbing xanthene PPGs were obtained [73]. It is crucial to emphasize the exploration of the photoactive properties of these photosensitive groups as promising photosensitizers for combined cancer therapy.

Our research group has also designed a conjugate with a photosensitive group that would operate simultaneously through both PDT and chemotherapy mechanisms. For the initiation of photolysis with long-wavelength light, we modified a well-known short-wavelength *o*-nitrobenzyl linker by incorporating a triple bond. This modification resulted in the creation of a unified conjugated system of porphyrin PS–linker, designed to undergo two-photon activation (Figure 20A, compound **37a,b**) [38]. As the therapeutic component, we employed the antimitotic agent *trans*-combretastatin A-4 **38**, which could undergo isomerization into the clinically active *cis*-isomer **2** upon exposure to both UV-A (λ = 365 nm) light and two-photon excitation [93].

Thus, the creation of such a hybrid conjugate could achieve minimal systemic toxicity and precise control over the processes of photorelease and drug activation. However, single-photon activation with UV-A light (λ = 365 nm), corresponding to the absorption maximum of the *o*-nitrobenzyl linker, did not lead to the desired photorelease, as did the use of other wavelengths (λ = 254 nm, 311 nm, white light) with varying intensity. At the same time, the fragment of conjugate **39,** which did not contain porphyrin, underwent slow photocleavage with the trans-combretastatin **38** release, which immediately isomerized into the cis-form **2** (Figure 20B). However, by the end of photorelease, phenanthrene derivative **40** became the main product. Additionally, calculations using the TD-DFT method for conjugates without hydrophilic groups revealed that, upon irradiation with light at a wavelength of λ = 365 nm, electronic transitions to three closely spaced in energy orbitals, LUMO, LUMO+1, and LUMO+2, are possible. Only on LUMO+2 is the electron density increased at the benzyl position of the linker, thus reducing the likelihood of photorelease in this case. Similar calculations for other possible types of linkages between porphyrin and *o*-nitrobenzyl linker showed that the use of the linker hindered photorelease.

R. Weinstein et al. combined both organic and metal–organic photocleavable groups in a single molecule [94]. They proposed using a porphyrin scaffold **41** as the metal–organic framework for the PPG (Figure 21).

Metalloporphyrin complexes are traditionally widely used in PDT, allowing for precise tuning of the photophysical parameters and bioavailability of PS. The *meso*-methyl position, involved in well-known organic PPGs such as coumarins, *o*-nitrobenzyl, and meso-methyl-BODIPY, was utilized as the photocleavable linkage. Photoactivation of synthesized meso-methylporphyrins **41**, including zinc, palladium, and copper complexes, was achieved with green (λ = 545 nm) and red (λ = 640 nm) light with the formation of alcohol **42**. The release of anticancer drugs indibulin and methotrexate occurred with high quantum yields of photorelease, confirmed by in vitro experiments. An important advantage of the created photocleavable system is the ability to incorporate four photosensitive linkages simultaneously in the *meso*-positions of the porphyrin, enabling the delivery of different drugs concurrently.

## 4. Activated PSs

One of the main challenges of PDT is the lack of selective accumulation of PSs in tumor tissues. To overcome this limitation, activatable PSs are being developed, which can be activated in tumor cells under the specific stimuli. Thus, during therapy, normal tissues retain their viability. One strategy for creating such a PS involves introducing certain fragments into its composition that act through various sensing mechanisms (Förster resonance energy transfer—FRET, photoinduced electron transfer—PET, and intramolecular charge transfer—ICT). The result of their influence on the PS is a change in the degree of ROS generation and fluorescence emission under light exposure (Table 3).

Since the therapeutic action of PSs is based on achieving efficient intersystem crossing responsible for Type 1 and Type 2 photoreactions, regulating possible competing processes of excited state deactivation leads to precise tuning of its therapeutic effect activation. The key idea in designing such activatable PSs lies in linking the PS and a quenching fragment, sensitive to the tumor microenvironment (acidic pH, hypoxia, excess of certain enzymes, and receptors) [95]. Thus, upon entering tumor cells, sensitive linkages are disrupted, and, as a result, energy transfer, charge transfer, or electron transfer processes cease, and the PS restores its therapeutic and fluorescent action.

**Table 3 pharmaceutics-16-00479-t003:** Comparison of FRET, PET, and ICT mechanisms.

	Requirements	Influence on Fluorescence and ROS Generation	Trigger	Reference
FRET	Overlap of donor emission and acceptor absorption spectra	Increase in acceptor fluorescence, quenching of donor fluorescence	Breaking or changing the bond between donor and acceptor	[96]
PET	The presence in the molecule of a donor and an acceptor	Full quenching	Changing pH, addition ions, carbohydrates, phosphates	[97]
ICT	The presence in the molecule of a donor and an acceptor, forming a dipole upon excitation	Fluorescence maximum shift, partial quenching	Changing pH or solvent polarity	[98]

Among the radiative processes that compete with ISC and impact the performance of PS, FRET is the most commonly encountered (Figure 22).

FRET occurs when there is an overlap between the donor’s emission spectrum (typically the PS) and the acceptor’s absorption spectrum (acting as a labile quencher), resulting in the quenching of the donor’s fluorescence and the activation of the acceptor’s fluorescence [99]. A prerequisite for FRET is the specific design of suitable donor and acceptor pairs, as well as the correct distance between them, usually less than 10 nm [100].

2,4-dinitrobenzenesulfonate is frequently used as a fluorescence quencher through FRET/PET, which is sensitive to glutathione [101]. Sulfonethers and sulfonamides are easily removed by thiol action, including GSH. Meanwhile, GSH is abundant in tumor cells and inside cells in general (~10 mM) compared to the intercellular space (~2 mM). Therefore, the action of GSH on the PS with a DNBS group selectively restores the photoactivity of the PS in tumor cells [102,103,104].

The research of J.-Y. Liu et al. is an example of implementing an activatable PS strategy. Here, two BODIPY photosensitizers **43**, one serving as a photosensitizer and the other as a quencher, are connected by a disulfide linker that is cleaved in the presence of the GSH (Figure 23) [105]. One BODIPY **44** fragment (λ_abs_ = 662 nm) acts as the PS, incorporating two iodine atoms into its core, effectively generating singlet oxygen (Φ_Δ_ = 0.3 in PBS with 0.05% Tween 80). The emission spectrum of **44** overlaps with the absorption maximum λ_abs_ = 705 nm of the second BODIPY fragment (quencher) **45**. The authors demonstrated that, in the bound state between the two BODIPY molecules, FRET occurs from PS **44** to quencher **45**. Thus, the fluorescence intensity of fragment **44** and singlet oxygen generation are inhibited. The presence of dimethylamino groups in the quencher **45** promotes ICT, resulting in weak fluorescence of this BODIPY fragment. However, in the presence of millimolar concentrations of glutathione, typical for tumor cells, the disulfide bond in **43** is cleaved, leading to the restoration of fluorescence and ROS generation. Upon light irradiation (λ = 670 nm, 20 mW/cm^2^, 2.4 J/cm^2^) of HeLa, A549, and H22 tumor cell lines incubated with conjugate **43**, high submicromolar toxicity was observed (IC_50 light_ = 0.67 mM for HeLa cells, 0.44 mM for A549 cells, and 0.48 mM for H22 cells). Moreover, conjugate **43** was non-toxic in the dark and, in the case of nontumoral HELF cells, it exhibited no toxicity either in the light or in the dark. Under laser irradiation of H22 tumor-bearing mice with conjugate **43** excellent results in inhibiting tumor growth were demonstrated compared to control groups without irradiation and non-cleavable conjugate.

Another strategy for creating activatable PSs involves the use of two-photon activation. In recent decades, two-photon PDT has been developed, which simultaneously uses two low-energy photons to reach the phototherapeutic window (650–900 nm), where the body tissues are most permeable to light. However, many porphyrinoid PSs have low two-photon absorption cross-sections. To overcome this limitation, electron transfer can be employed. For instance, FRET from a two-photon-absorbing donor to a PS that does not have a high two-photon absorption cross-section itself results in fluorescence emission in the long-wavelength region. In 2018, a photo-theranostic agent with strong absorption and fluorescence in the NIR region was created based on FRET [106,107,108]. Conjugates **46a,b** were formed with the two-photon-absorbing agent 2-acetyl-6-dimethylaminonaphthalene as the FRET donor and a non-metal or zinc tetraphenylporphyrin as the acceptor (Figure 24).

The morpholine fragment within the porphyrin PS facilitated targeting to lysosomes. Single-photon irradiation experiments with **46a,b** (λ = 490 nm) revealed high singlet oxygen quantum yields (Φ_Δ_ =0.57 for **46a**) and (Φ_Δ_ =0.66 for **46b**) and micromolar cytotoxicity upon irradiation of A549 cells. Two-photon irradiation of A549 cells with **46a,b** was performed using a femtosecond laser (λ = 740 nm, 115 mW, 80 MHz, 140 fs), resulting in strong red fluorescence in the cells within just 15 min. Additionally, lysosome and other membrane destruction in the cytoplasm were observed and, after 30 min of irradiation, morphological changes in the cell structure indicated cell death. In 2022, H. Zhou et al. presented conjugate **47**, consisting of a coumarin-containing two-photon FRET donor and a pyridine-containing coumarin acceptor [108]. Conjugate **47** could simultaneously generate superoxide anion radicals (type 1 photoreaction) and singlet oxygen (type 2 photoreaction) upon two-photon activation with λ = 840 nm light. It is important to note that type 1 photoreactions are particularly effective under tumor hypoxia, where molecular oxygen is present in limited amounts. In vivo studies of conjugate **47** on H22 tumor-bearing mice showed a reduction in tumor volume over a 14-day treatment period and good biocompatibility.

In 2020, R. A. Decréau et al. developed a FRET-activatable photosensitizer for dual Cherenkov radiation-induced near-infrared luminescence imaging and PDT [109]. Here, Cherenkov radiation energy (λ= 300–500 nm, from the β^+^ [^18^F]Fluorodeoxyglucose) was absorbed by a pyranine fluorophore with strong absorption in the range of λ = 250–450 nm. Through simultaneous TBET (through bond energy transfer) and FRET from the pyranine moiety to the phthalocyanine PS, emission of **48** was achieved in the near-infrared range (λ_fl_= 710 nm) (Figure 25). Consequently, the radiation source in the form of a radiopharmaceutical was introduced together with the conjugated photosensitizer **48** into tumor cells, allowing the therapeutic effect of PDT to extend beyond the depth of light penetration from an external source. The authors suggested that their strategy can detect more tumor areas and those pathological areas that were not completely removed can be treated with photosensitizers emitting from the radiopharmaceutical.

During the deactivation of the excited state of the photosensitizer, FRET, ICT, and PET compete with each other and with ISC, resulting in a reduction in the quantum yield of singlet oxygen. In addition, photoinduced electron transfer allows tuning the fluorescence mode of a photosensitizer from “off” to “on” and vice versa (Figure 26) [110]. Therefore, appropriately tuned PET enables the activation of fluorescence and the generation of singlet oxygen in tumor tissues under the influence of various stimuli such as acidic conditions, or the presence of certain ions, carbohydrates, or phosphates.

For example, in the study of X. Dong et al., PET was employed for selective detection of tumors and dual photodynamic and photothermal therapy [111]. The aza-BODIPY photosensitizer **49**, with absorption in the red spectral region, was modified by introducing a morpholine fragment that actively associates with lysosomes (Figure 27A). To control PET from the morpholine fragment on BODIPY, an acidic pH generated by the microenvironment of tumor cells was utilized. Under neutral conditions, the HOMO energy of morpholine significantly increases, thereby unblocking the PET pathway for conjugate **49**. This process competes with radiative/nonradiative transitions or ISC, effectively placing **49** in an “off” mode. However, in a slightly acidic environment, morpholine exists in a protonated form, lowering the HOMO energy level for **50**. This activation of pathways for radiative relaxation, vibrational relaxation, and ISC of **50** allows for the generation of singlet oxygen as well as a fluorescence response. To study the biological properties of the photosensitizer **49**, it was encapsulated in the amphiphilic polymer DSPE-mPEG2000. In in vitro experiments (HeLa cells), an excellent phototherapeutic effect of the photosensitizer was demonstrated with an IC_50 light_ = 10 µg/mL upon irradiation (λ = 730 nm, 3 min, 1.0 W/cm^2^). In vivo analysis for **49** confirmed that its pronounced phototherapeutic efficiency in an acidic environment is achieved through synergistic PDT/PTT simultaneously with good biocompatibility and rapid metabolic kinetics.

In 2021, Y. Zhao et al. achieved the regulation of PET in an acidic environment through the protonation of the diethylamine group of the BODIPY photosensitizer [112]. For its delivery to tumor cells, a targeted ligand—cyclic Arg-Gly-Asp peptide (cRGD)—was utilized, selectively binding to integrin ανβ3 receptors overexpressed by tumor cells [113]. At acidic pH created by lysosomes, the PET process was blocked, leading to the restoration of fluorescence and photodynamic activity of the protonated photosensitizer.

In order to enhance the selective activation of the photosensitizer, D. K. P. Ng et al. introduced dual activation of the photosensitizer using different tumor-specific stimuli [114]. Silicon phthalocyanine **51** was employed as the PS, connected to two ferrocene quenchers through pH-sensitive hydrazone and thiol-sensitive disulfide linkers (Figure 28). In a neutral environment, the asymmetric phthalocyanine **51**, due to PET from the ferrocene fragments, did not exhibit fluorescence and photoactive properties. However, in an acidic environment (pH 4.5–6.8) and in the presence of overexpressed dithiothreitol, the hydrazone and disulfide bonds were disrupted, blocking PET. Upon irradiation of the released photosensitizer **52** with red light (λ > 610 nm, 40 mW/cm^2^, 48 J/cm^2^), the fluorescence signal was amplified and singlet oxygen was generated. Remarkable results in inhibiting tumor growth upon irradiation with **51** were achieved in mice with HT29 human colorectal carcinoma. By improving the targeted phthalocyanine platform with the introduction of cathepsin-B and glutathione-sensitive fragments, dimeric and trimeric photosensitizing systems were created [115]. In addition to quenching with PET from the 2,4-dinitrobenzenesulfonate group, self-quenching of the phthalocyanine photosensitizers was observed due to their strong aggregation.

In the study by X. Peng et al., a fluorescein derivative exhibiting thermally activated delayed fluorescence was utilized for the bioimaging and photodynamic therapy of hypoxic tumors (Figure 29, compound **53**) [116]. The fluorescein-based photosensitizer **53** contained a *p*-nitrobenzyl moiety acting as a PET donor, thereby blocking fluorescence and singlet oxygen generation. However, in the presence of nitroreductase, which is abundant in hypoxic regions of the tumor, a cascade process was initiated. It led to the removal of the *p*-aminobenzyl fragment in **54**, regenerating strong fluorescence and singlet oxygen generation in compound **55**. When HeLa cells are incubated with the compound **53** in the presence of 10% oxygen, the photodynamic therapy efficiency (λ = 590 nm LED (16 mW/cm^2^)) was higher (IC_50 light_ = 6 μM) than under normoxic conditions (21% O_2_, IC_50 light_ > 20 μM). Moreover, the photoactivity of **53** under hypoxic conditions surpassed that of the PpIX photosensitizer (IC_50 light_ = 8 μM), attributed to its selective accumulation in lysosomes. Two-photon activation of conjugate **53** (λ = 890 nm, 35 GM) expands its potential applications for two-photon photodynamic therapy and treatment of deep-seated tumors.

Due to the presence of electron-donating and electron-accepting groups in the molecule, intramolecular charge transfer can occur during the transition from the S_0_ to the S_1_ state. This process involves the redistribution of electron density within the molecule, leading to the formation of dipoles and the creation of a charge transfer (CT) state. CT competes successfully with radiative relaxation and ISC, resulting in a significant shift in the fluorescence spectrum or a reduction in its intensity, as well as the inhibition of singlet oxygen generation (Figure 30) [98]. Acidity or solvent polarity change increase the energy of CT, enhancing the probability of the fluorescent relaxation and ISC pathways, and, consequently, singlet oxygen generation.

Several studies have investigated the influence of ICT on the ability of BODIPY molecules to generate singlet oxygen [117,118]. BODIPYs exhibit bright fluorescence, making them widely used as fluorescent labels but limiting their photosensitizing ability. To enhance the quantum yield of the triplet state and the efficiency of ISC, heavy atoms are introduced into the core of BODIPY [119]. However, this approach may lead to an increase in dark cytotoxicity [120]. Another strategy to increase the quantum yield of singlet oxygen involves regulating ICT in orthogonal BODIPY-dimers depending on the solvent polarity. It has been observed that, in such dimers, the quantum yield of singlet oxygen increases in relatively nonpolar solvents. Conversely, in polar solvents, efficient charge transfer from the 8-substituted BODIPY (donor) to the 2-substituted BODIPY (acceptor) reduces the probability of ISC, as well as the quantum yield of singlet oxygen [121]. Thus, by regulating ICT, it is possible to block the photoactivity of BODIPY-dimers, reducing unwanted damage to healthy cells during therapy. For instance, X. Chen et al. created a combined agent **56** for dual PDT–chemotherapy activated by cathepsin B in tumor cells (Figure 31) [122]. The orthogonal BODIPY-dimer **57** contained an electron-donating amino group responsible for ICT, making the photosensitizer actively generate singlet oxygen. The authors proposed to block ICT by binding the amino group to a cathepsin-sensitive peptide, thereby reducing ROS generation. The photosensitizer was then linked with a cRGD-modified PEG chain and used as nanocarriers **56** to load a hydrophobic anticancer agent, 10-hydroxycamptothecin. After selective binding of the cRGD peptide to the integrin receptor αvβ3, overexpressed in certain tumors, cathepsin B-catalyzed regeneration of the amino group of the photosensitizer **57** occurred. This activated ICT and triggered the photosensitizing activity of the BODIPY **57**, as well as released the chemotherapeutic drug.

The combined effect of PDT and chemotherapy of nanoparticles **56** was observed during the incubation of 4T1 mouse breast cancer cells, inducing apoptosis upon irradiation. In 3D models of tumor growth, the nanoparticles **56** penetrated into the tumor and inhibited their growth upon laser irradiation. However, short-wavelength blue light (λ = 488 nm) was used, limiting the application of PS **56** to superficial tumor cells.

In the study by J. Zhao et al., a BODIPY diad was developed, where one of the fragments is pH-sensitive [118]. BODIPY, absorbing green light, acted as an ICT donor for another diiodostirylbene BODIPY (acceptor), absorbing red light and containing a dimethylamino group. Upon protonation of the amino group, charge transfer from the donor to the acceptor was realized, leading to a tenfold enhancement in singlet oxygen generation (from Φ_Δ_ = 7% to Φ_Δ_ = 74%). In this case, ICT resulted in an increase in the excited triplet state’s lifetime of BODIPY.

In 2017, Zh. Xie et al. developed the photosensitizer BODIPY **58**, activated by two sensing mechanisms simultaneously (PET and ICT) (Figure 32) [123]. The trimethoxyphenyl group was introduced into the *meso*-position of BODIPY **58,** providing PET. A similar mechanism of fluorescence quenching was also demonstrated with a phenolic group in the *meso*-position of BODIPY and a pH-sensitive linker on HeLa cells [124]. The nitrovinyl group in the 5-position of BODIPY **58** acted as a quencher through the ICT. However, in the presence of biothiols, abundant in tumor cells, they undergo Michael addition to the nitrovinyl group, inhibiting ICT and PET. Simultaneously, the ability of PS **59** to fluoresce was regenerated (a 30-fold increase in fluorescence quantum yield when excited at λ = 530 nm) and exhibited a photodynamic effect. This process was verified on the HeLa tumor cell line upon glutathione addition.

Conjugate **58** also demonstrated a high ratio of dark-to-light cytotoxicity on HeLa tumor cells (PI = IC_50 dark_/IC_50 light_ = 353) and HepG2 (IC_50 dark_/IC_50 light_ = 153), while no significant toxicity was observed in normal BEAS-2B cells (green LED (515–525 nm)).

## 5. Bioorthogonal Delivery

Until recently, the challenge of selective PS delivery was addressed through targeting specific cells by directly attaching a targeting moiety to the PS or by creating nanosized/liposomal delivery systems incorporating targeting ligands [125,126]. However, a true advancement in selective drug delivery has been achieved with bioorthogonal chemistry, awarded the Nobel Prize in 2022 [127,128,129,130,131]. The biocompatibility achieved through the chemical and biological inertness of bioorthogonal reactions has elevated the selective accumulation of drugs in pathological formations to a new level, as demonstrated in vivo.

For precise control of drug delivery, it is necessary to introduce a targeting moiety with a bioorthogonal group and then add the drug with another suitable bioorthogonal group (Figure 33). In this case, the drug, while circulating in the bloodstream, will predominantly react with the bioorthogonal label of the targeting moiety at its localization site. In the case of PDT, the selectivity of PS accumulation can be significantly enhanced [132]. Furthermore, drug delivery has been improved by introducing the PS in its “deactivated” form, i.e., it was unable to exhibit a fluorescent response and generate singlet oxygen. The bioorthogonal reaction not only ensured the specificity of PS binding to the target but also activated the PS, enabling selective treatment [133].

Within this review, examples of photosensitizing systems utilizing the two most common bioorthogonal reactions—1,3-dipolar cycloaddition of azides and alkynes (click reaction) and the inverse electron demand Diels–Alder reaction (iEDDA)—are discussed (Table 4).

Considerable attention has been drawn to the copper-catalyzed [3+2]-cycloaddition reaction of azide to alkyne, which occurs with high-rate constants k = 10–10^2^ M^−1^s^−1^. In 2021, a two-photon-activatable ruthenium polypyridyl complex **60** as a photosensitizer was delivered into triple-negative breast cancer cells (MDA-MB-231) using click chemistry (Figure 34) [134]. The researchers utilized the technology of metabolic labeling of glycans with a bioorthogonal chemical reporter such as azide [135]. In the first step, N-azidoacetylmannosamine (ManNAz) **61** binds to the plasma membrane of tumor cells and integrates through the glycan biosynthesis into various glycoconjugates. In the next step, the ruthenium photosensitizer **60**, modified with a triple bond, binds to glycans containing compound **61** through click chemistry. The resulting compound **62** generated singlet oxygen (Φ_Δ_ = 0.80) with a low dose of two-photon irradiation (λ = 810 nm, 6 J/cm^2^, 5 min). Additionally, ruthenium complex **62** with bioorthogonal delivery demonstrated high cytotoxicity (IC_50 light_ = 10.6 ± 0.87 μM). However, its dark-to-light cytotoxicity ratio (PI = IC_50 dark_/IC_50 light_) was 18.9. It is worth noting that, in non-tumor cells (MCF-10A) incubated with **60** with or without **61**, no significant toxicity was observed. Thus, ruthenium complex **60** with bioorthogonal delivery demonstrates selective action in MDA-MB-231 tumor cells.

R. Yang et al. used N-azidoacetylmannosamine **61** for the delivery of a chlorin-*e*_6_ derivative, which, together with the dibenzocyclooctyne bioorthogonal label, was attached to polymeric nanoparticles [136]. Bioorthogonal delivery of the PS was demonstrated on three tumor cell lines: 4T1, HeLa, and MCF-7. At acidic pH, the nanoparticles released the chlorin-*e*_6_ derivative, producing a fluorescent signal. Moreover, in vivo studies of nanoparticles with nude mice bearing the 4T1 tumor showed inhibition of tumor growth.

It is known that, when activating a PS with red light, healthy muscle tissues near bladder cancer cells can be damaged [137]. To address this issue, photosensitizers with absorption at shorter wavelengths are needed. Y. You et al. used an azide-containing rhodamine B **63**, with an absorption maximum at λ = 531 nm, for bladder cancer cell bioimaging and subsequent therapy (Figure 35) [138]. For efficient singlet oxygen generation, the authors employed energy transfer (FRET) from rhodamine B **63** to a phthalocyanine photosensitizer **64**. The azide group of rhodamine B **63** participated in a click reaction with the cyclooctyne fragment of phthalocyanine **64**. PDT selectivity was achieved by controlling the irradiation wavelength. Specifically, the short-wavelength hv_1_ (λ = 531 nm) cannot activate phthalocyanine **64**, which remains in normal tissues, but activates rhodamine B, linked to phthalocyanine in compound **65**, in tumor cells. As a result of FRET, the phthalocyanine fragment in **65** generated singlet oxygen. Even at low concentrations of **64** and rhodamine **63** (M = 5 · 10^−7^ M (λ = 531 nm)), a rapid reaction between them (60 min in human bladder carcinoma T-24 cells) reduced cell survival by 90%. Meanwhile, rhodamine B **63** and phthalocyanine **64** separately showed minimal toxicity under light irradiation. The combination of **63** and **64** in the dark also proved non-toxic to T-24 cells.

In 2021 a new strategy to enhance the specificity of a photosensitizer for tumor cells was proposed by B. Liu et al. Using a bioorthogonal reaction, the authors suggested synthesizing a photosensitizer inside pathological cells [139]. For this purpose, precursors containing a triple bond **66** and an azido group **67** were incorporated into the nanoscale MOF-199 (Figure 36). Due to the presence of quaternized amino groups, compounds **66** and **67** accumulated in the mitochondria. When exposed to Cu(I) ions as catalysts, precursors **66** and **67** underwent a reaction with each other. Within the framework of MOF-199, Cu(II) ions were initially incorporated to generate Cu(I) ions, with the reduction of Cu(II) by glutathione resulting in Cu(I). Interestingly, the photosensitizer **68** formed through the click reaction exhibited aggregation-induced emission (AIE). In vitro studies of the photoactive system on the HeLa tumor cell line with glutathione hyperexpression demonstrated high phototoxicity (white light, IC_50_ = 10 μM). At the same time, the toxicity of the photoactive system in 3T3 cells with low glutathione expression was negligible. The developed strategy of “synthesis of a photosensitizer activated by cancer cells” provided PDT with visual monitoring, including in vivo (zebrafish with HeLa tumors).

The solid tumor microenvironment poses a variety of biological barriers, including tumor hypoxia, reduced pH, and other adverse factors that reduce the effectiveness of PDT [140]. Moreover, most photosensitizers initiate type II photochemical reactions, transforming triplet oxygen into singlet oxygen, intensifying local hypoxia [141]. Therefore, photosensitizers aim to be introduced into oxygen-rich tumor vasculature to efficiently generate ROS. Simultaneously, hypoxia-activatable prodrugs (HAPs) are employed to combat hypoxia [142]. In pursuit of a combined effect of PDT and chemotherapy, Y. Yuan et al. developed a pH-sensitive nanoscale delivery system for a chlorin-based photosensitizer and HAP using bioorthogonal chemistry (Figure 37) [143]. Previously, the group led by A. J. MacRobert demonstrated the potential for targeted delivery of a chlorin-*e*_6_ derivative modified with benzocyclooctyne via SPAAC with azido-TAT peptide, specifically targeting endosomal membranes [144].

When the nanocarrier based on poly(2-azepane ethyl methacrylate) entered an acidic environment, it was protonated. This led to the activation of the dibenzocyclooctyne group (DBCO) within the nanocarrier, which reacted with the azide group of dendritic polyamidoamine (PAMAM) modified with the PR104A HAP. Through the click reaction, enhanced accumulation of HAP and chlorin-*e*_6_, which is part of the nanocarrier, in the tumor was achieved. Under laser irradiation with a wavelength of λ = 660 nm in normoxic conditions, chlorin-*e*_6_ was released from the drug depot and could efficiently generate ROS. In turn, ROS formation led to increased hypoxia and the disruption of the ROS-sensitive thioketal linker between HAP and PAMAM. Consequently, released HAP penetrated hypoxic regions, where the hypoxia-activatable therapy with PR104A destroyed tumor cells. Based on the results of in vitro experiments (4T1 cells) and in vivo studies (4T1 orthotopic tumor-bearing BALB/c mice), the presented combined system with laser illumination (λ = 660 nm) demonstrated enhancement of the combined PDT and hypoxia-activated chemotherapy effect.

As of now, the highest reaction rates among bioorthogonal reactions have been achieved using the inverse electron-demand Diels–Alder reaction (k = 1–10^6^ M^−1^s^−1^). In this [4+2]-cycloaddition reaction, an electron-withdrawing diene (such as 1,2,4,5-tetrazine **69**) and an electron-donating dienophile (alkene **70** or alkyne) react through the π4s + π2s scheme, yielding a highly strained bicyclic intermediate compound **71** (Figure 38). The adduct **71** undergoes retro-Diels–Alder reaction with nitrogen release, forming the corresponding 4,5-dihydropyridazine **72**, which either isomerizes into the respective 1,4-dihydro counterparts **73** or oxidizes to produce the pyridazine product **74** [145]. Moreover, unlike the click reaction, where catalytic amounts of metals (copper or ruthenium) are required, the iEDDA reaction does not necessitate the presence of a catalyst.

With iEDDA reaction, R. Weissleder et al. achieved photocontrol of the BODIPY photosensitizer activity [146]. Upon combining BODIPY and tetrazine, exclusively efficient energy transfer and fluorescence quenching via the TBET were observed. When tetrazine **75** reacted with *trans*-cyclooctenol **76** (TCO), the energy transfer ceased and the fluorescence intensity of BODIPY increased by over a thousand times (Figure 39). O. Vázquez et al. first applied the diiodo-BODIPY-tetrazine system for bioorthogonal activation of PDT within cell nuclei. The DNA was modified with the dienophile 5-vinyl-2’-deoxyuridine [147]. In this work, the authors also provided computational justification for energy transfer from the excited S_2_ state of BODIPY to the S_1_ state of tetrazine through FRET. To achieve PDT activation in specific organelles, a series of BODIPY/tetrazine conjugates were obtained, separated by a long linker based on various peptide fragments [148]. In this series, effective quenching of BODIPY fluorescence was not achieved due to increased spatial separation of the energy donor and acceptor. However, compound **77** (Figure 39) exhibited pronounced photodynamic effects (IC_50 light_ = 0.096 ± 0.003 μM for HeLa cells, λ = 525 nm LED (69.4 ± 0.6 W/m^2^ for 160 s)) upon irradiation, coupled with negligible dark toxicity (cell viability at 4 μM concentration of **77** was around 100%). Additionally, for targeting the BODIPY/tetrazine system to tumor cells, *trans*-cyclooctene linked to biotin was used. It is known that biotin receptors are abundant in many types of tumors, such as HeLa [149].

D. K. P. Ng et al. developed a bioorthogonal system that employed two bioorthogonal reactions simultaneously—iEDDA and click chemistry—for precise activation of PDT in cells. The photosensitizer **78** contained distyrylBODIPY with tetrazine and alkyne fragments (Figure 39) [150]. The expansion of the conjugated π-system by condensing aromatic aldehydes and the BODIPY core to form distyrylBODIPY allowed the use of red light (λ > 610 nm) for PS activation. TCO-labeled GE11 peptide bound to EGFR on the surface of A431 tumor cells and was used for the iEDDA reaction. Simultaneously, metabolic glycoengineering with tetraacetylated N-azidoacetyl-D-mannosamine **61** was used for the click reaction. Having both tetrazine and alkyne fragments, distyrylBODIPY **78** could specifically bind to cancer cells through both iEDDA and click reactions. This dual bioorthogonal ligation enhances phototherapeutic effect of conjugate. The phototoxic effect of **78** was demonstrated in vitro (IC_50_~10 μM) and in vivo (λ > 610 nm, 18 mW/cm^2^, 30 min) on tumor-bearing nude mice. Importantly, without irradiation or prior administration of TCO-labeled peptide and **61**, distyrylBODIPY **78** was non-toxic (IC_50_~100 μM). It is worth noting that, in the case of distyrylBODIPY **78** (λ_fl_ = 710 nm, Φ_fl_ = 0.02), the tetrazine fragment could no longer effectively quench the fluorescence of the photosensitizer. Due to the lack of spectral overlap between the emission maxima in the red or near-infrared region of BODIPY and the absorption of tetrazine, energy transfer between them is not achieved [151].

The introduction of a carboxylate group into the *meso*-position of distyrylBODIPY is known to result in improved fluorescent properties. To “mask” the photoactivity of the photosensitizer, the same authors proposed introducing a self-immolative fragment via the carboxyl group [152]. For targeted delivery of the photosensitizer **79** to tumor cells, a [4+1] cycloaddition reaction of isocyanide with tetrazine was utilized (Figure 40). To achieve this, a 3-isocyanopropyl group was incorporated into the structure of distyrylBODIPY. The tetrazine fragment was conjugated with tumor-specific ligands (galactose derivative and GE11 peptide specific to EGFR). The [4+1] cycloaddition reaction was accompanied by N_2_ removal, forming a pyrazolimine intermediate **80**. After hydrolysis, the resulting aldehyde **81** underwent β-elimination to generate the corresponding phenol **82**, which then self-removed, releasing the photoactive carboxydistyrylBODIPY **83**. In vitro studies (λ > 610 nm, 23 mW/cm^2^, 28 J/cm^2^, cell lines with varying levels of EGFR expression) and in vivo experiments (A431 tumor-bearing nude mice) demonstrated rapid kinetics and high tumor specificity for compound **79**.

In 2024, D. Ye et al. employed the bioorthogonal iEDDA reaction for multimodal synergistic PDT of tumors with fluorescent and MRI monitoring [153]. As the photosensitizer, they chose silicon phthalocyanine **84** incorporated into nanoparticles, modified with tetrazine fragments (Figure 41). To alleviate hypoxia in tumor cells, the authors used a low-molecular-weight carbonic anhydrase (CA) inhibitor **85**, conjugated with tetrazine. *Trans*-cyclooctene **86**, linked with ^68^Ga and a fluorescent dye sensitive to alkaline phosphatase, served as the tetrazine partner in iEDDA. Self-assembly of multiple TCO fragments from **86** into nanoparticles occurred through dephosphorylation, penetrating the tumor membranes and providing fluorescent and MRI signals. Subsequently, the PS **84** and the CA inhibitor **85** were introduced in nanoparticles. Due to tetrazine fragments **84** and **85,** rapid binding occurred via iEDDA with a large number of TCOs, forming bulk microparticles. The authors demonstrated that such bulk microparticles exhibit extended retention time in the tumor. The next step was the irradiation with λ = 808 nm light (0.33 W/cm^2^) for 10 min, activating the PS **84.** The efficiency of PDT increased due to the inhibition of CA activity and reduction of hypoxia (HeLa-IC_50 light normoxia_ = 3.52 ± 0.55 μM and HeLa-IC_50 light hypoxia_ = 2.55 ± 0.30 μM). The authors showed that subcutaneous HeLa tumors in mice could be completely eradicated without observed tumor recurrence. Moreover, the feasibility of controlling tumor PDT in live mice with high sensitivity and spatial resolution was demonstrated using dual NIR fluorescence signals (λ = 710 and λ = 780 nm) and magnetic resonance imaging signals.

A green indocyanine derivative, a clinically approved PS active in the near-infrared range, was also selectively delivered to tumors through the iEDDA reaction [154]. The hydrophobic photosensitizer was encapsulated in liposomal particles, rendering it hydrophilic, resulting in increased circulation time in the bloodstream and retention in tumor cells. After the liposomes were disrupted inside the cells, the hydrophobic PS associated with the mitochondria. Subsequent laser irradiation with λ = 808 nm light achieved a combined effect of PDT and PTT.

Tetrazines, as dienophile partners in the iEDDA reaction, exhibit some drawbacks. The most reactive tetrazines are unstable in the cellular environment for an extended period of time [155]. Additionally, they may display nonspecific reactivity towards strong nucleophiles [156]. Consequently, hydrolytically more stable tetrazine precursors, known as dihydrotetrazines, have been developed to provide more precise spatio-temporal control over the iEDDA reaction upon their oxidation into corresponding tetrazines. The oxidative conversion of dihydrotetrazine to tetrazine was achieved by J. M. Fox et al. using red light and methylene blue photocatalysts [157], as well as a silicon-derived rhodamine photocatalyst [158]. Another approach for photoactivation of «dihydrotetrazine-to-tetrazine» conversion was proposed by N. K. Devaraj et al. Dihydrotetrazine **87** was protected by a photocleavable group (Figure 42) [159]. Various PPGs were explored, including 1-(2-nitrophenyl)ethyl (λ_abs_ = 405 nm), 6-nitropiperonyl methyl (λ_abs_ = 425 nm), and diethylaminocoumarin (λ_abs_ = 450 nm). It was hypothesized that, upon light irradiation, the photolabile group is cleaved, forming a nitrogen-centered anion **88**, which, in the presence of oxygen, is oxidized to tetrazine **89**. Furthermore, the protected dihydrotetrazine was suggested to be enzymatically activated in tumor cells [160]. In our opinion, a promising avenue for converting a stable dihydrotetrazine into a highly reactive tetrazine could involve using red- or near-infrared-absorbing groups as PPGs (such as *meso*-methyl-BODIPY derivatives or cyanine dyes).

Ongoing research suggests the potential for identifying novel, faster, and more efficient bioorthogonal reactions [161,162]. The improvement of such reactions is likely to focus on the creation of bioorthogonal reactions resembling natural bonds, such as amide, esters, and phosphodiethers. In addition to selectively delivering therapeutic molecules to tumor cells, the concept of delivering and subsequently releasing active drugs (PSs) through a bioorthogonal reaction seems appealing (Figure 43) [163].

To mask the activity of PS, primarily composed of amino and hydroxyl groups, it is proposed to conjugate them with a dienophile, such as vinyl or TCO-group. Subsequent iEDDA reaction with the tetrazine fragment bound to the tumor cell leads to PS elimination, forming either aromatic derivative **85** or **86** [164]. The “click and release” concept has already been tested for masking a fluorescent agent, enabling the visualization of DNA [165]. However, so far, no attempts have been made to use this strategy for the targeted release of photosensitizers during PDT. It is evident that the high speed and specificity of bioorthogonal chemistry could greatly enhance the selectivity of PDT and lead to the development of corresponding prodrugs.

## 6. Conclusions

PDT has become widely used in treating various types of cancer due to its non-invasiveness, targeted action, and immune-modulating properties. However, certain limitations, such as non-selective accumulation in target tissues leading to significant toxicity in healthy tissues, limited effectiveness in the hypoxic conditions of tumor cells, and inadequate tissue penetration, hinder its integration as a primary treatment method for cancer alongside surgery, chemotherapy, and radiation therapy.

To overcome these limitations, activatable photosensitizers have been rapidly developed in recent years. This approach involves deactivating the photosensitizer beforehand and then activating it within the target tissue using various stimuli, allowing for precise PDT tuning. Light is particularly notable among the stimuli for switching the PS from an “off” to an “on” state, as using the necessary light source for therapy also serves to switch the PS states, providing high levels of spatial and temporal control over PS action. For example, light-induced formation of singlet oxygen and other ROS triggers the decomposition of sensitive-to-singlet-oxygen aminoacrylate and thioketal linkers, widely used for drug delivery and combined therapy. Modeling PS structures with donor and acceptor fragment adjustment enables the use of FRET and PET, and ICT upon PS photoexcitation to deactivate its fluorescence and ROS-generating properties, thereby reducing the phototoxicity of healthy tissues during and after therapy.

It is important to note that combining quenching processes and light-sensitive fragments in a conjugated PS significantly enhances PDT selectivity. For instance, in a PS/chemotherapeutic conjugate, deactivating the PS in advance allows for the subsequent release of two distinct drugs—the photosensitizer and the chemotherapy agent—upon exposure to light. Furthermore, the concept of PACT, based on masking the drug’s active site with a photosensitive group, has been reconsidered in recent years for implementing combined (PDT + chemotherapy) therapy. Successful examples with ruthenium complexes and meso-methyl-BODIPY as both photosensitive groups and photosensitizers have been reported. However, a balance between high quantum yield of photorelease reaction and ROS generation needs to be found, as these processes may compete with each other. It is anticipated that near-infrared-absorbing cyanine and xanthene photoactivatable groups will also be investigated as photosensitizers in the context of PDT.

The success in using photoactivatable PSs largely hinges on creating multifunctional molecules capable of efficiently absorbing light and undergoing photochemical reactions. Therefore, we anticipate that numerous upcoming studies will focus on designing photosensitizing systems. New and effective photosensitizers will be developed to absorb light in the NIR and be tailored to specific types of activation (PET, FRET, ICT, TBET, etc.). An interesting perspective is the application of two-photon activation. However, it is important to note that not every PS can be activated in this way.

The focus of research on PDT in recent years has shifted significantly towards the use of bioorthogonal reactions, driven by their excellent biocompatibility and high reaction rates. Among such reactions, the [3+2]-cycloaddition of azide to alkyne and the inverse electron-demand Diels–Alder reaction, primarily due to their readily available substrates and high-rate constants, have led to taking PS delivery to a new level of selectivity. Moreover, successful PS activation through bioorthogonal reaction has been achieved, not only binding it to tumor cells but also halting energy or electron transfer processes. Interesting examples include the combination of metabolic labeling and bioorthogonal reactions to enhance PDT effectiveness. For instance, incorporating an alkyne-containing bioorthogonal handle into a single PS molecule enables reactions with azide-tagged glycoproteins inside tumor cells. Additionally, an additional bioorthogonal handle in the form of tetrazine was utilized for selective binding to a peptide vector. This resulted in a platform with two independent levels of selectivity, promising the development of next-generation PDT drugs.

It is assumed that intensive research on discovering new bioorthogonal reactions will enable controlled PS release and the creation of chemical bonds characteristic of living organisms, creating a photosensitizing system inside the cell. Bioorthogonal drug release is a rapidly advancing field. Significant efforts have been dedicated to utilizing this technology for delivering conventional chemotherapy agents. We anticipate that this approach will also be extensively applied to deliver PS, thereby enhancing the targeted efficacy of PDT.

We hope that many of the approaches discussed for creating activatable photosensitizers will be translated into clinical PDT, significantly increasing humanity’s chances in the fight against cancer.

## Figures and Tables

**Figure 1 pharmaceutics-16-00479-f001:**
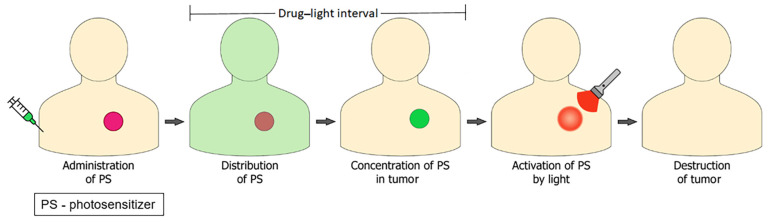
Treatment protocol for PDT of cancer.

**Figure 2 pharmaceutics-16-00479-f002:**
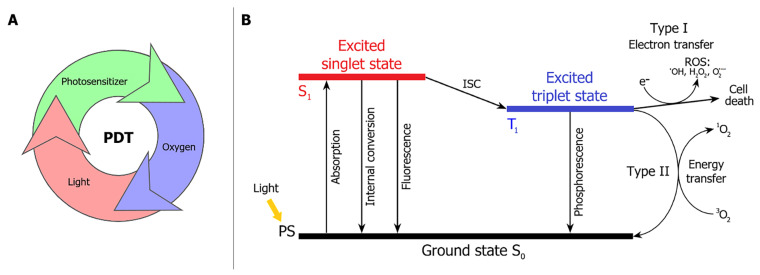
(**A**) PDT main components; (**B**) Jablonski diagram.

**Figure 3 pharmaceutics-16-00479-f003:**
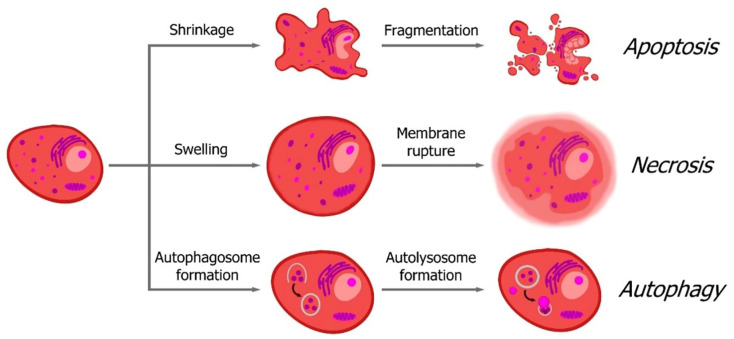
Cell death pathways induced by photodynamic therapy.

**Figure 4 pharmaceutics-16-00479-f004:**
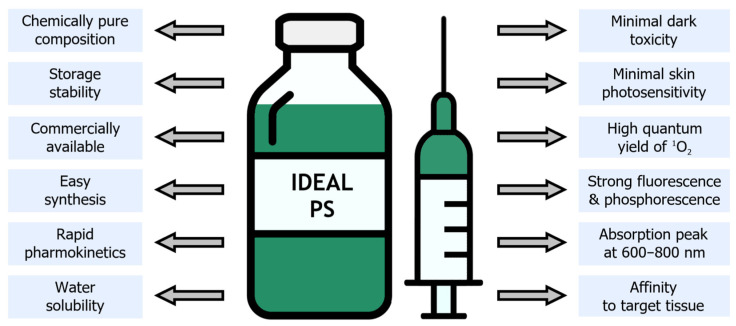
Characteristics of an ideal photosensitizer.

**Figure 5 pharmaceutics-16-00479-f005:**
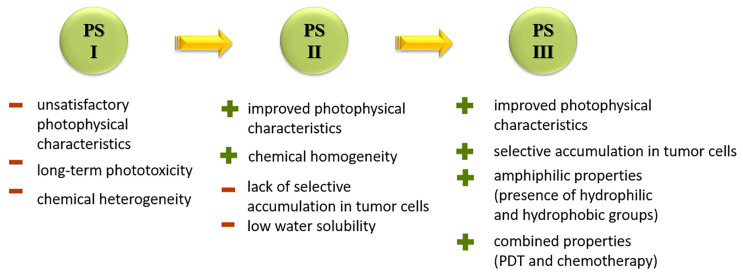
Comparison of three generations of photosensitizers.

**Figure 6 pharmaceutics-16-00479-f006:**
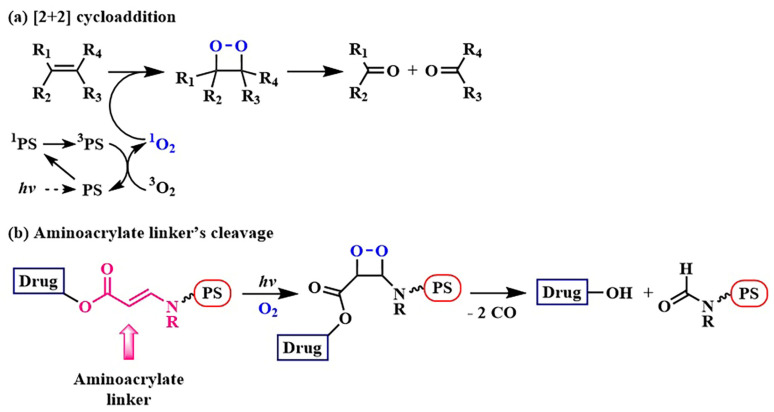
(**a**) [2+2]-cycloaddition of singlet oxygen to an alkene. (**b**) Design of aminoacrylate-containing conjugate and release of a drug upon activation during photodynamic therapy.

**Figure 7 pharmaceutics-16-00479-f007:**
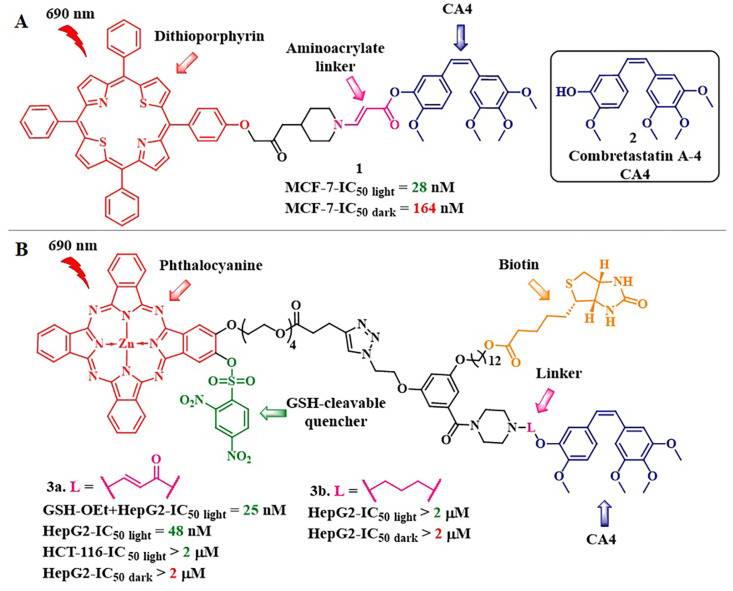
(**A**) Structure and cytotoxicity of dithioporphyrin-CA4 conjugate **1** with the β-aminoacrylate linker; (**B**) Structure and cytotoxicity of phthalocyanine-CA4 conjugate **2** with the β-aminoacrylate linker.

**Figure 8 pharmaceutics-16-00479-f008:**
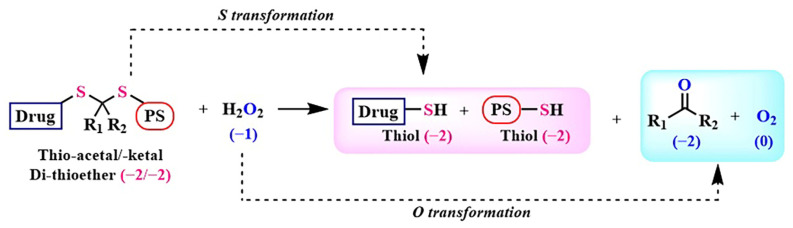
Thioacetal/-ketal transformations in the presence of ROS. The numbers represent the oxidation states of the chemical elements highlighted in the corresponding color.

**Figure 9 pharmaceutics-16-00479-f009:**
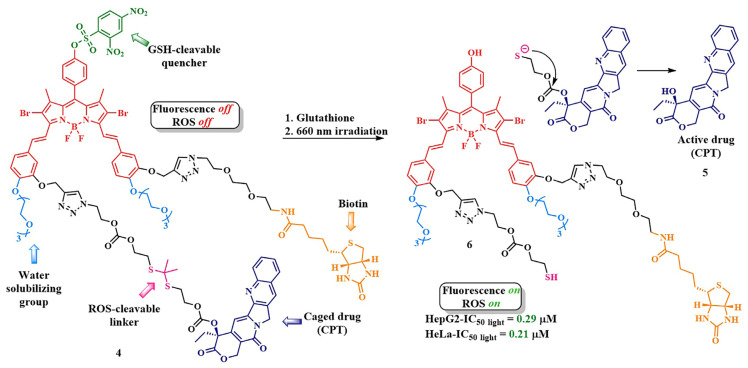
Camptothecin release upon destruction of the ROS-sensitive thioketal linker in conjugate **4**.

**Figure 10 pharmaceutics-16-00479-f010:**
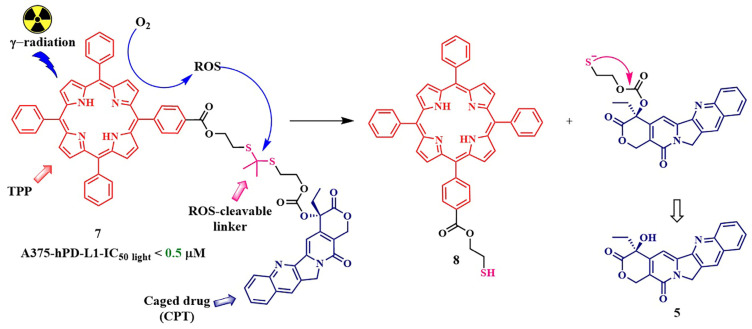
The structure and operational mechanism of conjugate **7**, activated by Cherenkov radiation.

**Figure 11 pharmaceutics-16-00479-f011:**
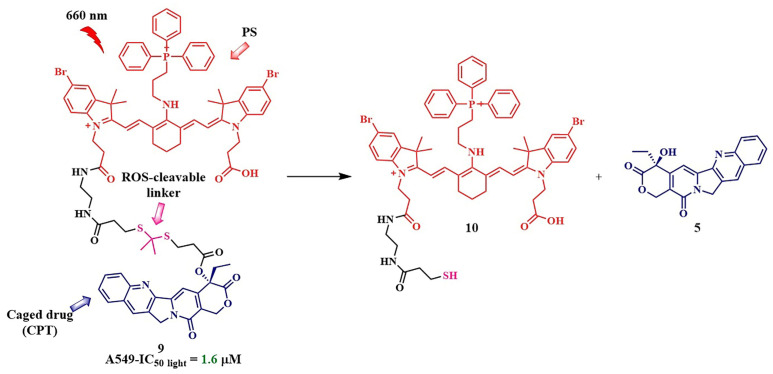
Camptothecin release upon destruction of the ROS-sensitive thioketal linker in conjugate **9.**

**Figure 12 pharmaceutics-16-00479-f012:**
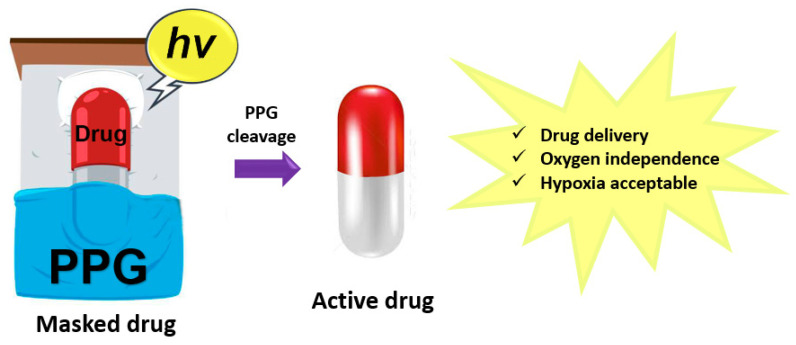
Photoactivated chemotherapy concept.

**Figure 13 pharmaceutics-16-00479-f013:**
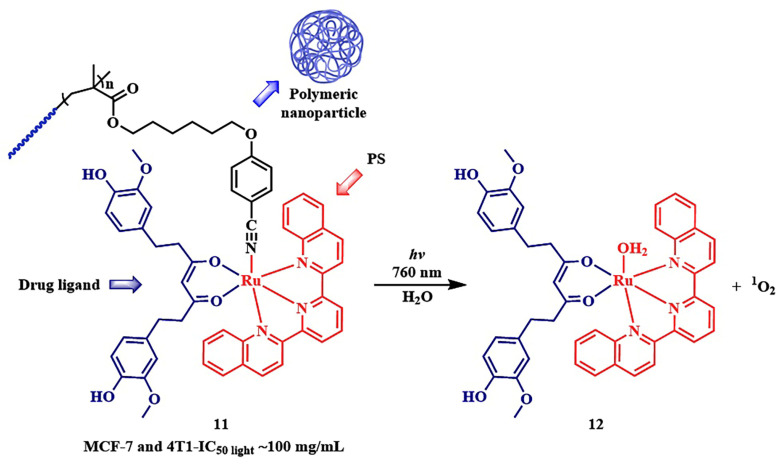
Photoinduced release of Ru-tetrahydrocurcumin complex **12** from compound **11.**

**Figure 14 pharmaceutics-16-00479-f014:**
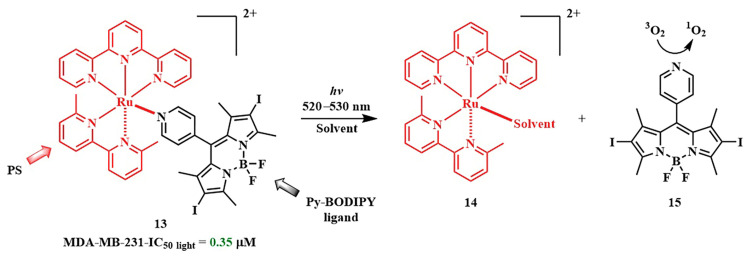
Photoinduced release of BODIPY **15** from compound **13.**

**Figure 15 pharmaceutics-16-00479-f015:**
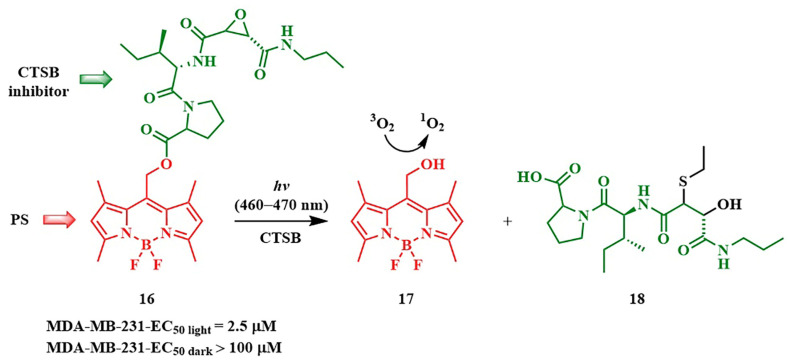
Photoinduced release of cathepsin B (CTSB) inhibitor **18** from conjugated PS **16.**

**Figure 16 pharmaceutics-16-00479-f016:**
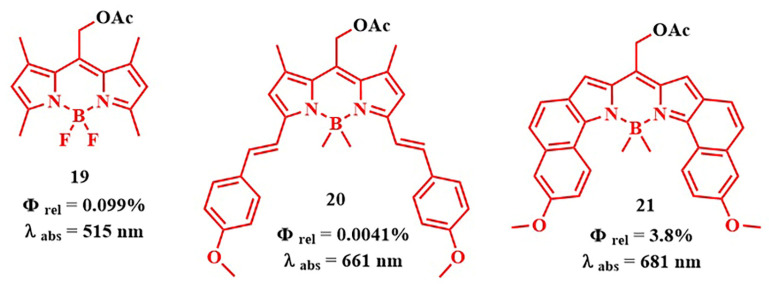
Structures of BODIPY PPGs.

**Figure 17 pharmaceutics-16-00479-f017:**
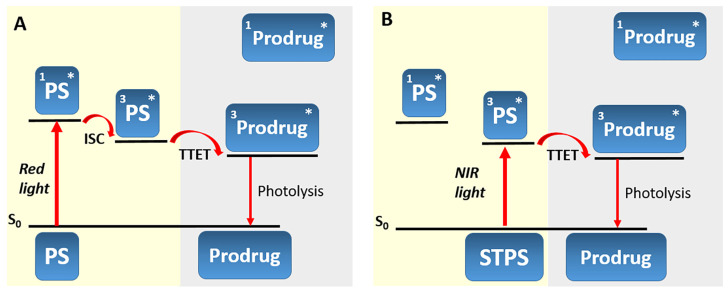
(**A**): Upconversion-like photolysis (**B**): NIR-light-triggered upconversion-like photolysis.

**Figure 18 pharmaceutics-16-00479-f018:**
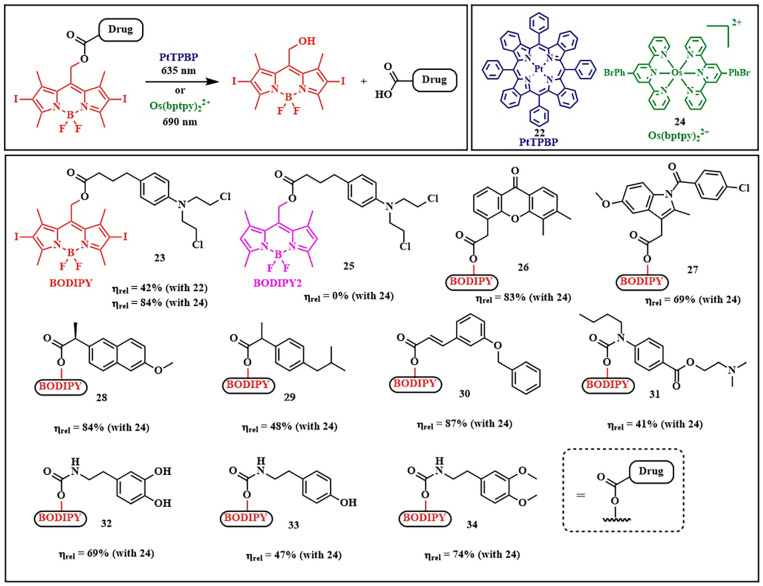
Upconversion-like photolysis systems.

**Figure 19 pharmaceutics-16-00479-f019:**
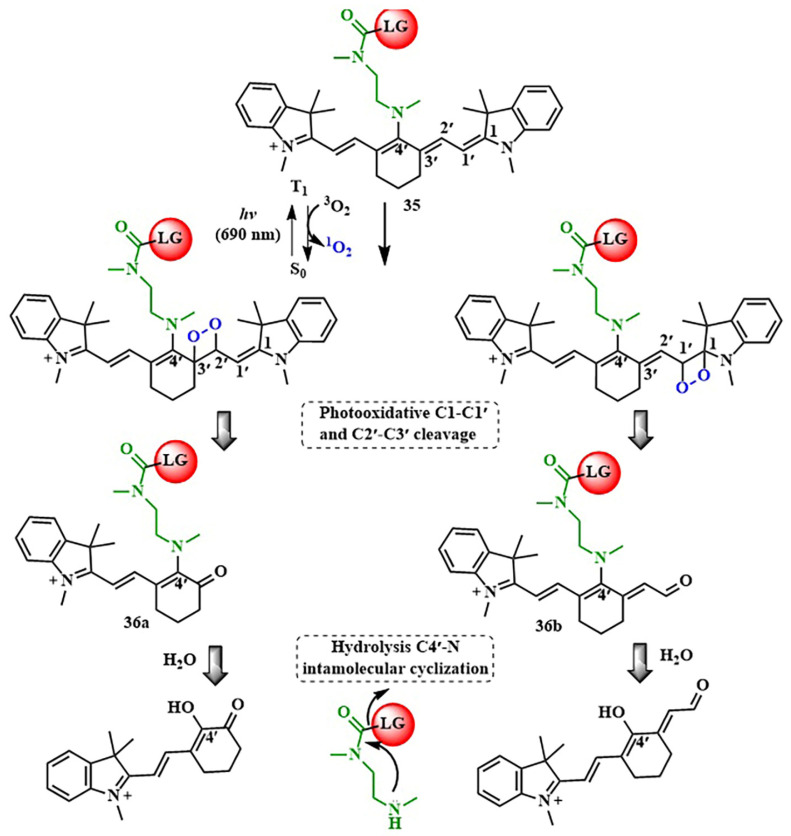
Uncaging reaction sequence of C4′-dialkylamine-substituted heptamethine cyanines **35**.

**Figure 20 pharmaceutics-16-00479-f020:**
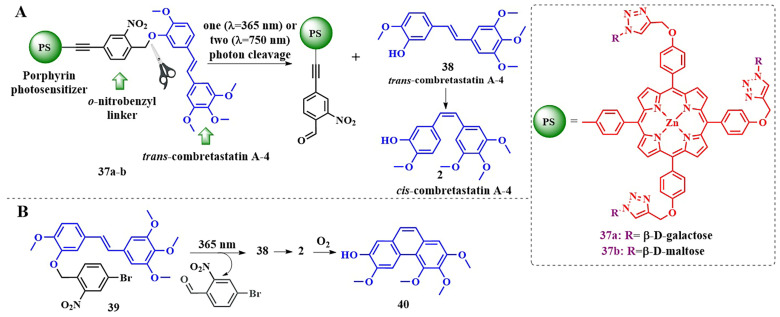
(**A**) Proposed photocleavage of conjugates **37a,b** under UV-A or two-photon irradiation. (**B**) Proposed mechanism of phenanthrene-type product **40** formation from *o*-nitrobenzyl linker **39** under UV-A light.

**Figure 21 pharmaceutics-16-00479-f021:**
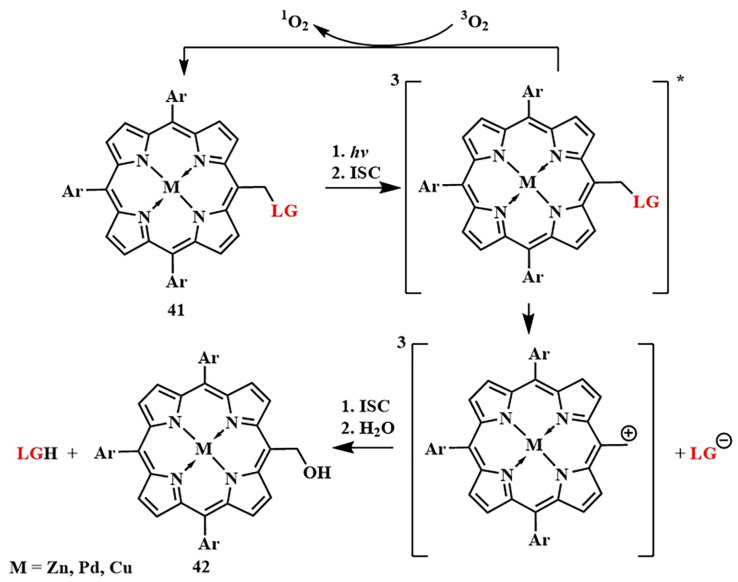
Proposed photorelease mechanism for porphyrin PPGs.

**Figure 22 pharmaceutics-16-00479-f022:**
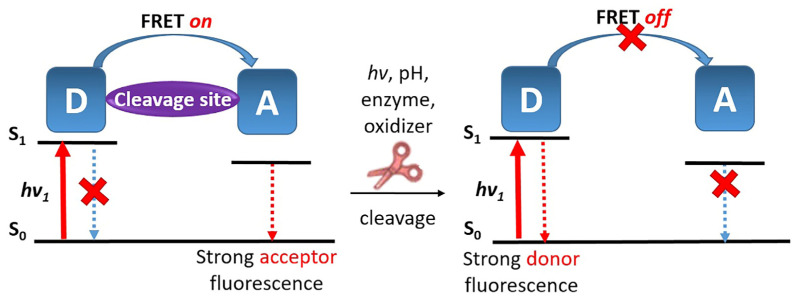
Schematic representation of FRET (D—donor, A—acceptor).

**Figure 23 pharmaceutics-16-00479-f023:**
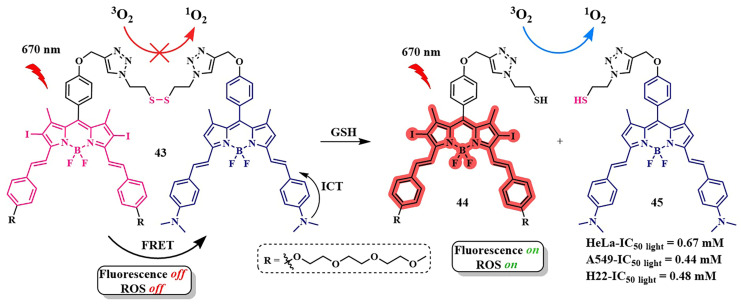
Mechanism of action of GSH-sensitive BODIPY-dimer **43**.

**Figure 24 pharmaceutics-16-00479-f024:**
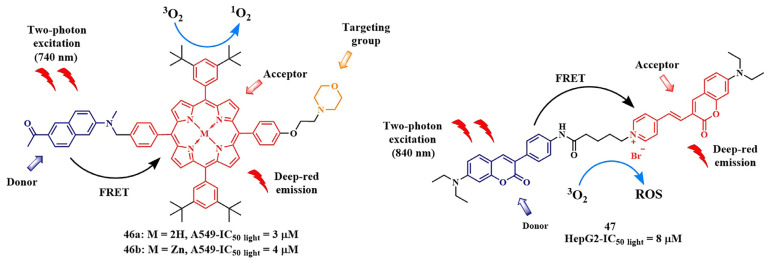
Two-photon excited FRET systems **46a,b** and **47.**

**Figure 25 pharmaceutics-16-00479-f025:**
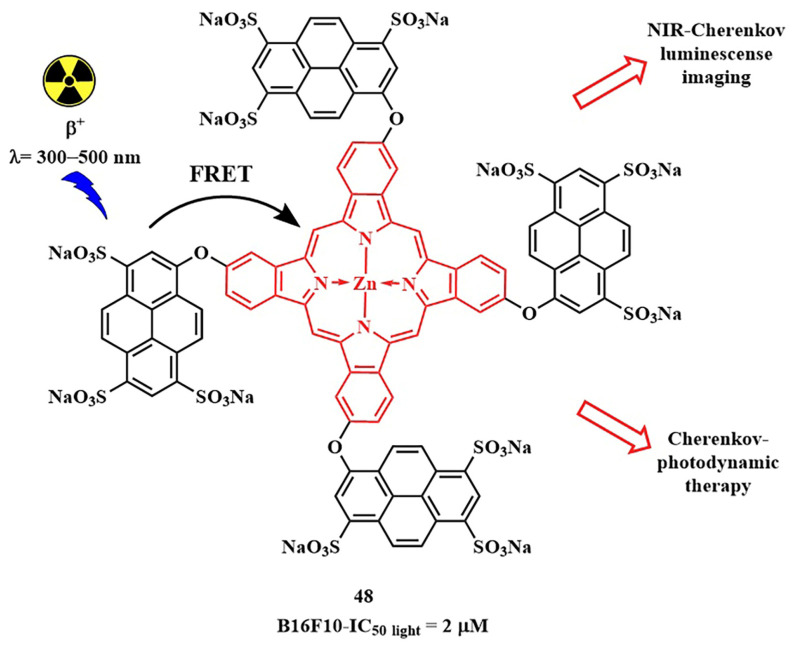
Structure of conjugate **48** activated by a Cherenkov radiation.

**Figure 26 pharmaceutics-16-00479-f026:**
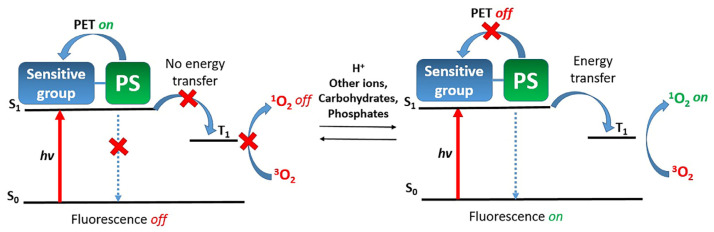
Schematic representation of PET.

**Figure 27 pharmaceutics-16-00479-f027:**
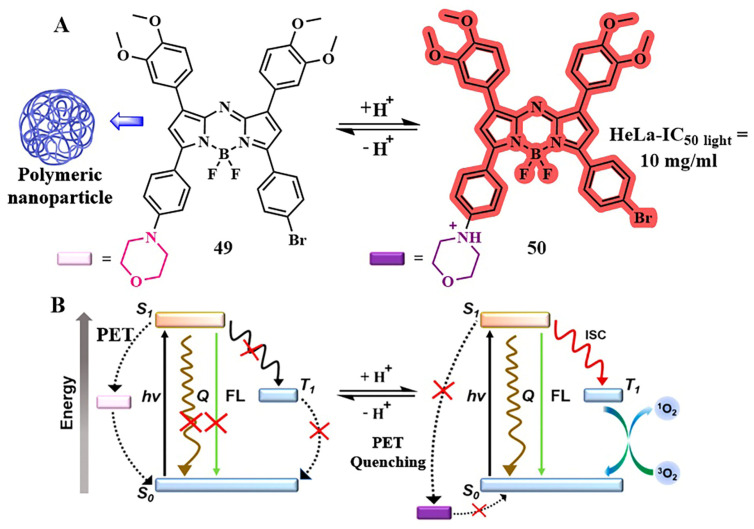
(**A**) Structure of conjugate **49**. (**B**) Adapted Jablonski diagram for conjugate **49.**

**Figure 28 pharmaceutics-16-00479-f028:**
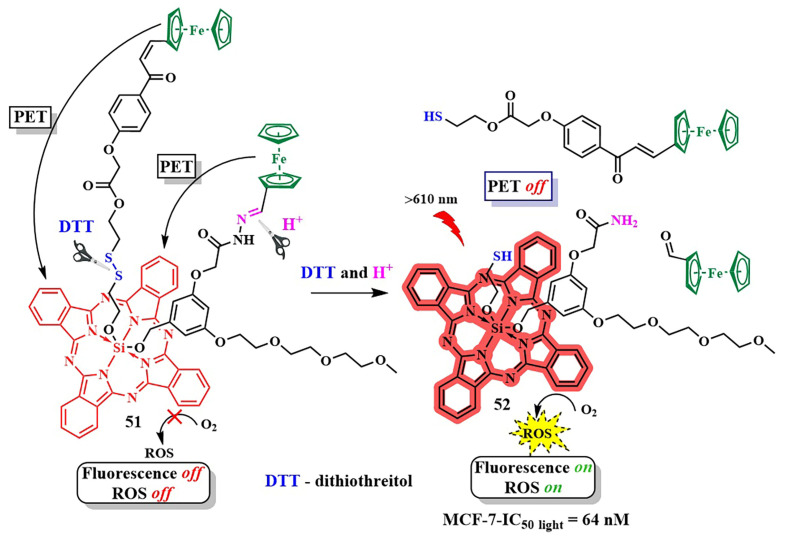
Structure of silicone phthalocyanine conjugate **51** with two ferrocene quenchers.

**Figure 29 pharmaceutics-16-00479-f029:**
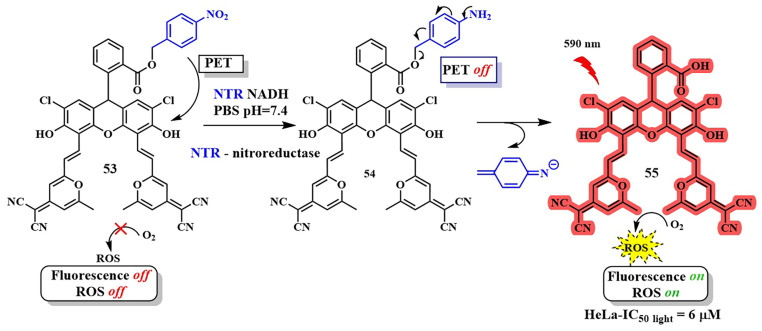
Nitroreductase-sensitive fluorescein derivative **53.**

**Figure 30 pharmaceutics-16-00479-f030:**
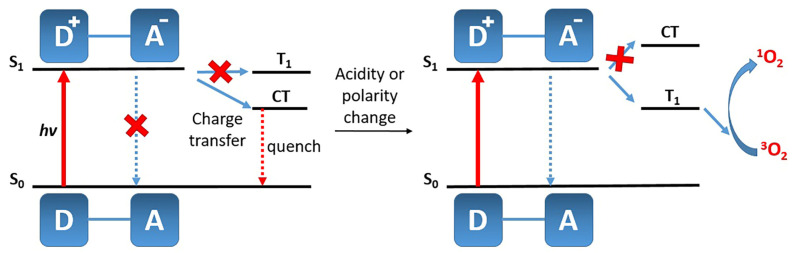
Schematic representation of ICT (D—donor, A—acceptor).

**Figure 31 pharmaceutics-16-00479-f031:**
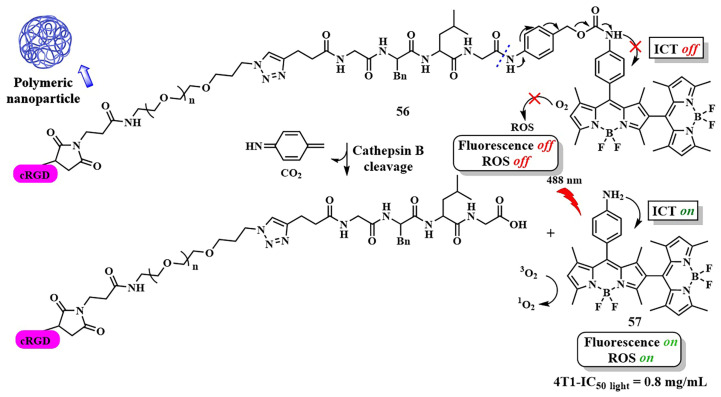
Cathepsin B-sensitive orthogonal BODIPY-dimer **56.**

**Figure 32 pharmaceutics-16-00479-f032:**
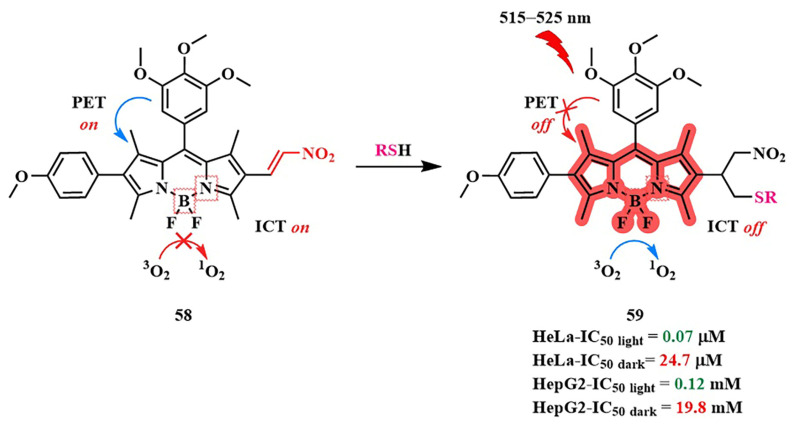
GSH-sensitive BODIPY photosensitizer **58.**

**Figure 33 pharmaceutics-16-00479-f033:**
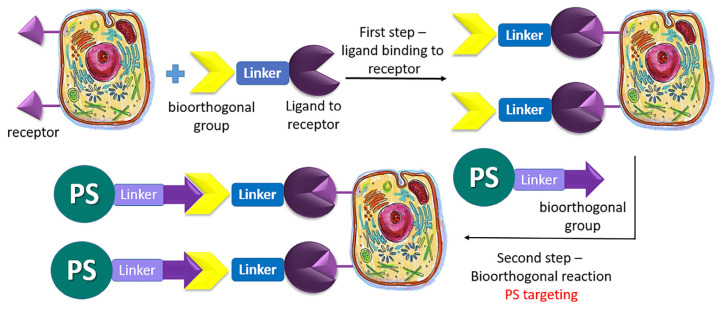
Schematic representation of bioorthogonal delivery of a photosensitizer to a tumor cell.

**Figure 34 pharmaceutics-16-00479-f034:**
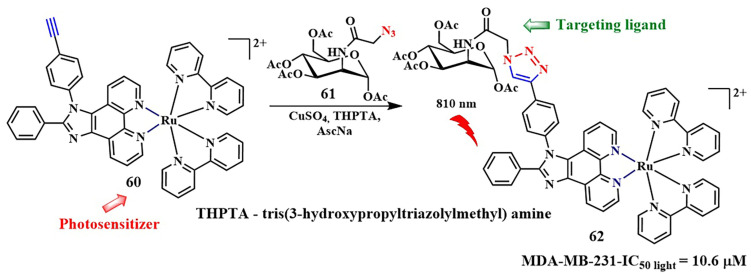
Bioorthogonal delivery of Ru-complex **60.**

**Figure 35 pharmaceutics-16-00479-f035:**
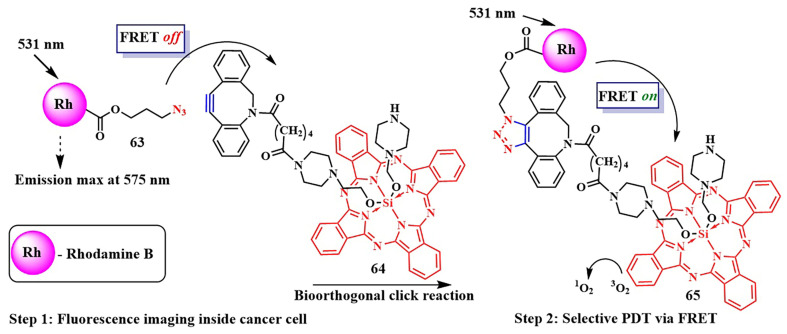
Click reaction between rhodamine B **63** and phthalocyanine **64.**

**Figure 36 pharmaceutics-16-00479-f036:**
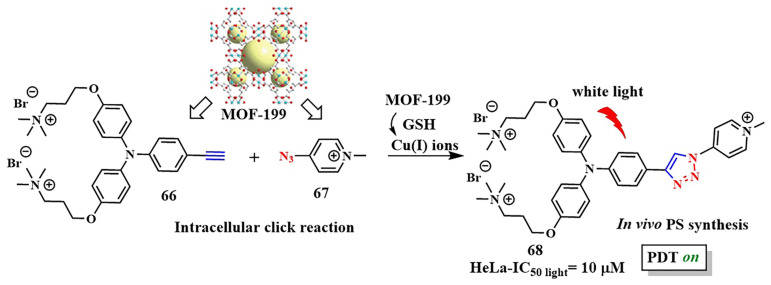
Intracellular PS **68**—formation via click reaction.

**Figure 37 pharmaceutics-16-00479-f037:**
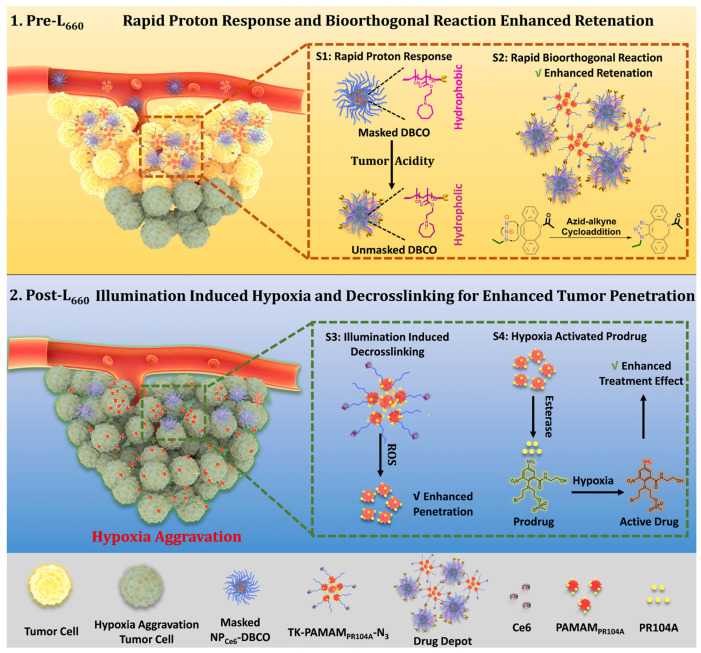
pH-sensitive nanoscale delivery system for a chlorin-*e*_6_-based photosensitizer and HAP. Reprinted with permission from ref. [143].

**Figure 38 pharmaceutics-16-00479-f038:**
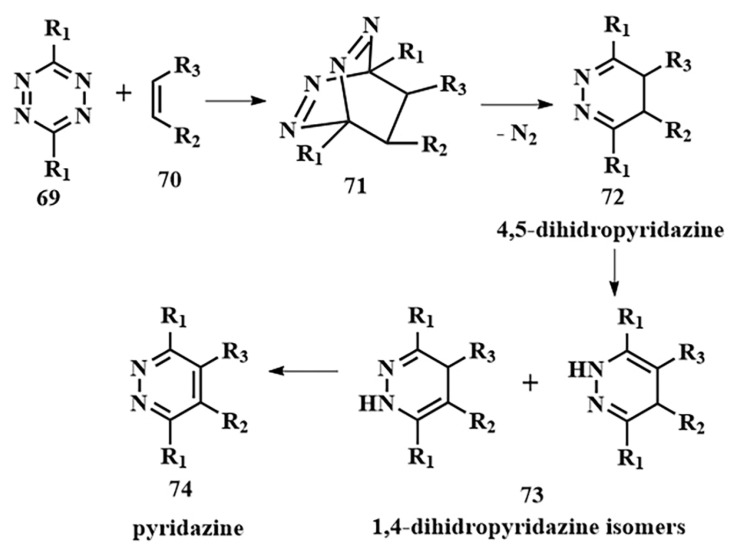
Inverse electron demand Diels–Alder reaction mechanism.

**Figure 39 pharmaceutics-16-00479-f039:**
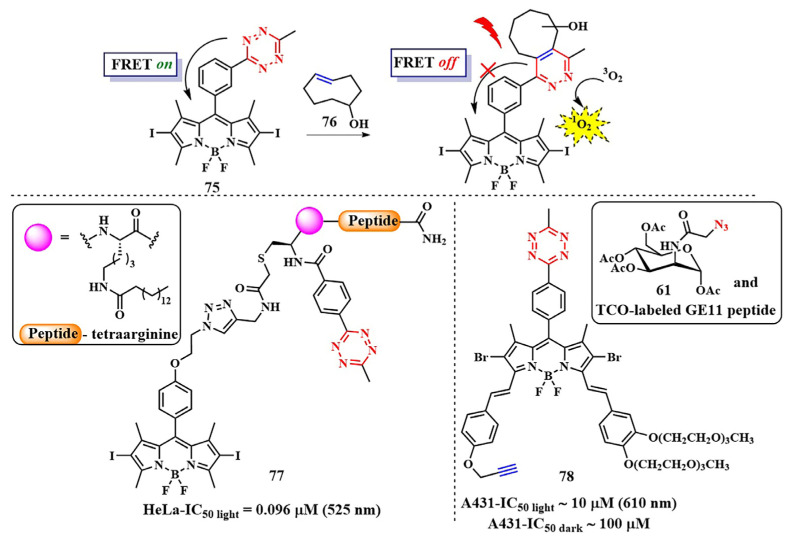
BODIPY/tetrazine PS biorthogonal delivery.

**Figure 40 pharmaceutics-16-00479-f040:**
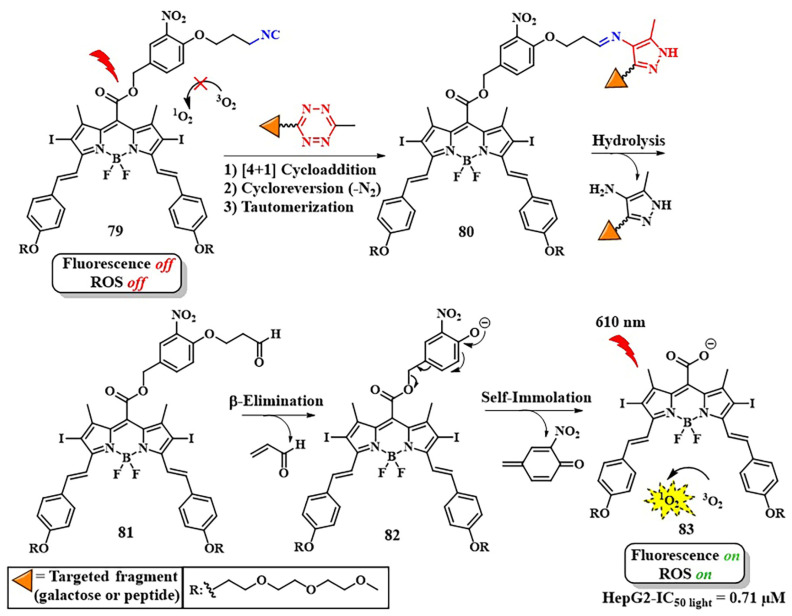
A tetrazine-responsive isonitrile-caged BODIPY photosensitizer **79.**

**Figure 41 pharmaceutics-16-00479-f041:**
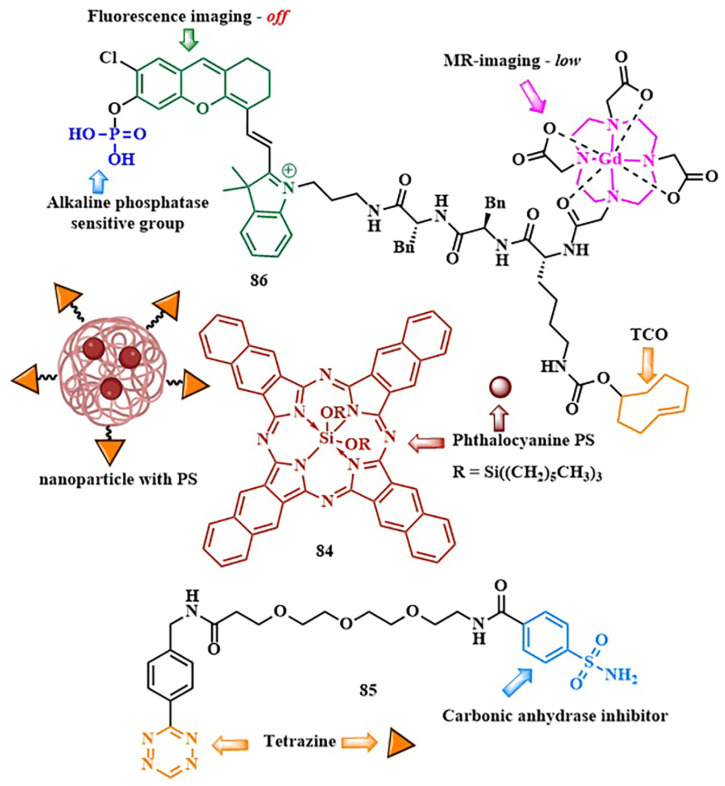
Theranostic system for pretargeted PDT.

**Figure 42 pharmaceutics-16-00479-f042:**
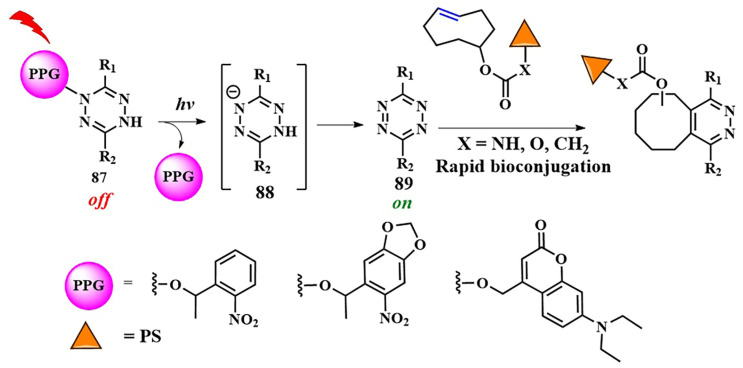
Light-activated dihydrotetrazine–tetrazine transformation and subsequent bioorthogonal reaction.

**Figure 43 pharmaceutics-16-00479-f043:**
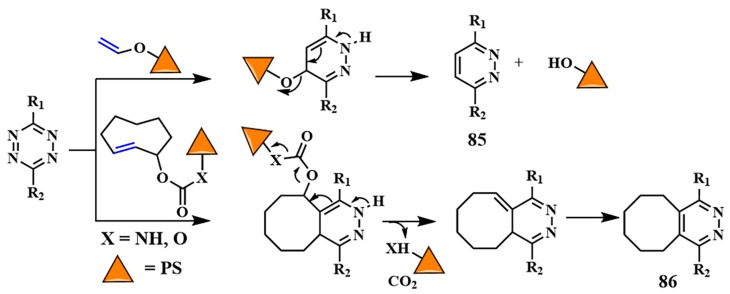
«Click and release» strategy for amine- and alcohol-containing PSs.

**Table 1 pharmaceutics-16-00479-t001:** Clinically used photosensitizers.

PSs	Commercial Name	λ_max._ (nm)	λ_max._·10^−3^ (M^−1^·cm^−1^)	Application	Refs.
First generation					
HpD (ether form) 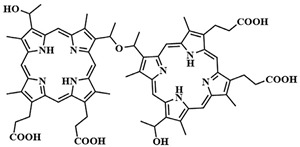	Photofrin	630	3	Lung, bladder, esophagus, cervical, brain, gastrointestinal cancer	[12]
Second generation					
Verteporfin 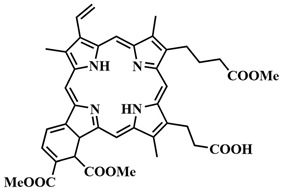	Visudyne	690	34	Skin, pancreas AMD, basal cell carcinoma	[6,8,12]
ALA-induced protoporphyrin IX 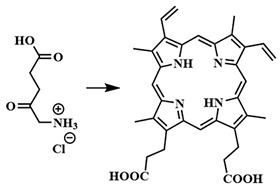	Ameluz	630	-	Actinic keratosis and basal cell carcinoma	[12]
	AlaCare	630	-	Actinic keratosis	[12]
	Levulan	635	5	Actinic keratosis	[12]
2-(1-hexyloxyethyl)-2-devinyl pyropheophorbide-*a*, HPPH 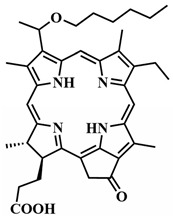	Photochlor	665	47.5	Lung, esophagus, head and neck cancer	[12]
*m*-tetra(hydroxyphenyl)chlorin 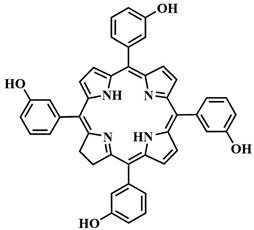	Foscan	652	30	Prostate, bronchus, pancreas, head and neck cancer	[12,13]
Chlorin-*e*_6_ derivatives 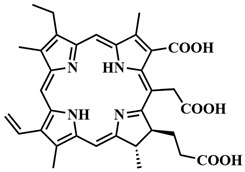	Photoditazine	668	48	Skin, breast, lung, prostate	[25]
	Photolon	665	50	Skin, breast, nasopharyngeal sarcoma	[12,13]
	Radachlorin	662	34.2	Skin, lung, brain	[6,12]
Talaporfin 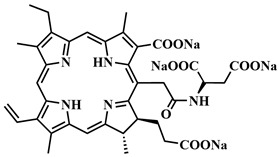	Laserphyrin	664	40	Lung, esophagus, brain, liver, colon	[6,12,24]
Redaporfin 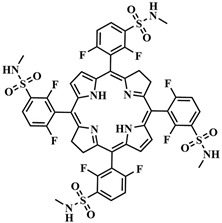	LUZ11	749	140	Biliary tract, head and neck cancer	[12,13,14]
Tin ethyl etiopurpurin 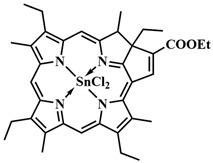	Purlytin	660	28	Breast, skin, prostate, Kaposi’s sarcoma	[8,14]
Lutetium texaphyrin 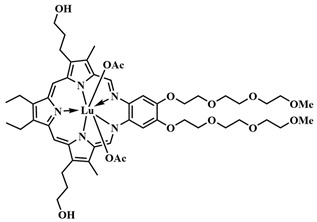	LUTRIN	732	42	Cervical, prostate, brain, AMD	[14,25]
Padeliporfin 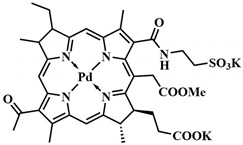	TOOKAD, WST11	762	110	Prostate	[13,14]
Zinc phthalocyanine 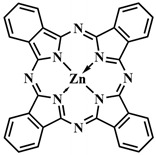	CGP 55847	670	200	Squamous cell carcinoma of upper aerodigestive tract	[13,14]
Tetrasulfonated chloroaluminum phthalocyanine 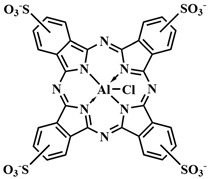	Photosens	675	200	Skin, breast, lung, cervical, larynx, head and neck cancer, liver and gastrointestinal cancer	[14,25]
Silicon phthalocyanine 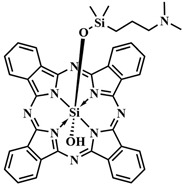	Pc4	675	200	Cutaneous neoplasms	[14,15]

**Table 2 pharmaceutics-16-00479-t002:** Comparison of red and near-IR absorbing photocleavable groups.

Photoprotective Group (LG-leaving Group)	Wavelength Range, nm	Release Quantum Yield (Ф_r_), %	Applicability to PDT	Reference
Ru-complex 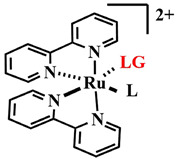	350–760	1·10^−2^–22	Yes	[70]
BODIPY 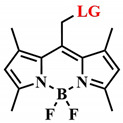	514–709	1·10^−3^–3.8	Yes	[71]
Cyanine 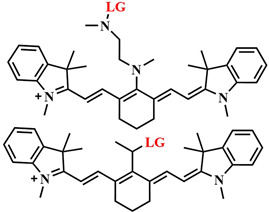	640–817	Up to 14	No	[72]
Xantene 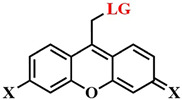	540–640	Up to 18	No	[73]

**Table 4 pharmaceutics-16-00479-t004:** Comparison of the click reaction and the inverse electron demand Diels–Alder reaction.

	Click Reaction	iEDDA
**Rate constant, k (M^−1^s^−1^)**	10–10^2^	1–10^6^
**Bioorthogonal handles**	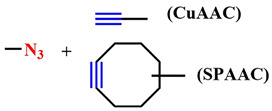	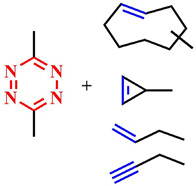
**Catalyst**	yes—Cu (I), except for the SPAAC	no
**In vivo application**	yes	yes
**Limitations**	requirement for toxic copper (I) ions or limitation by the structure of a hindered cyclic alkyne	need for precise tetrazine design due to effects on reaction rates, limited stability of tetrazines in water

## Data Availability

Not applicable.

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
