# Peer review of "Enhancing Precision in Photodynamic Therapy: Innovations in Light-Driven and Bioorthogonal Activation"

_pharmaceutics, 2024, doi:10.3390/pharmaceutics16040479_

Round 1
Reviewer 1 Report
Comments and Suggestions for Authors
Natalia S. Kuzmina and co-authors focused the enhancing precision in photodynamic therapy and reviewed the light-driven and bioorthogonal activation-related works. This review is well-organized and introduced the three generations of photosensitizers. After careful evaluation, it is suitable for publishment after addressing the following suggestions.
Comments on the Quality of English Language1) In lines 6-7, the e-mail address is duplicated and not necessary.
2) Please double check all the abbreviations in the whole manuscript, the full names of photosensitizer (PS), rug-light interval (DLI), reactive oxygen species (ROS), and photodynamic therapy (PDT) showed up many times in the whole manuscript.
3) In line 120, it should be 1st generation PSs, not 1nd.
4) In line 455 for Figure 16, the caption is better for Structures of …
5) Please update the style of reference 89.
Author Response
Dear Reviewer,
Thank you for your efforts in reviewing our manuscript and valuable feedback. Responding to the comments below, we revised the manuscript.
1) In lines 6-7, the e-mail address is duplicated and not necessary.
Corrected
2) Please double check all the abbreviations in the whole manuscript, the full names of photosensitizer (PS), rug-light interval (DLI), reactive oxygen species (ROS), and photodynamic therapy (PDT) showed up many times in the whole manuscript.
Corrected
3) In line 120, it should be 1st generation PSs, not 1nd.
Corrected
4) In line 455 for Figure 16, the caption is better for Structures of …
Corrected
5) Please update the style of reference 89.
Corrected
Reviewer 2 Report
Comments and Suggestions for Authors
Your study “Enhancing Precision in Photodynamic Therapy: Innovations in Light-Driven and Bioorthogonal Activation” is quite interesting and collect some useful information about new photosensitizers’ modifications. It is good illustrated by Jablonski’s diagrams and figures.
Please apply following corrections:
1. Introduction: not only dermatology: gynecology, dentistry, etc. Please add some information.
2. Figurer 4 – Strong fluorescence and “phosphorescence” both of these phenomena are useful in photodynamic diagnosis.
3. There is no information about delta-aminolaevulinic acid which is a precursor of proper PS and it is widely used in PDT.
Best regards,
Reviewer.
Author Response
Dear Reviewer,
Thank you for your efforts in reviewing our manuscript and valuable feedback. Responding to the comments below, we revised the manuscript.
1. Introduction: not only dermatology: gynecology, dentistry, etc. Please add some information.
Necessary information was added
2. Figure 4 – Strong fluorescence and “phosphorescence” both of these phenomena are useful in photodynamic diagnosis.
Necessary information was added
3. There is no information about delta-aminolaevulinic acid which is a precursor of proper PS and it is widely used in PDT.
Necessary information was added
Reviewer 3 Report
Comments and Suggestions for Authors
I read this review carefully, which presents the latest innovations in PDT with in light-driven and bioorthogonal activation.
The manuscript is very pleasant to read and well documented.
A few (minor) comments to further improve this article:
1- a table summarising all the acronyms would be welcome
2- Similarly, acronyms should be explained in all figure legends.
3- page 9 line 190: there is an extra dash at the beginning of the line
4- page 9 lines 211-226: this paragraph is not very clear, could the authors reword it?
5- page 10, line 225: did the authors check that the effect was indeed synergistic in the strict mathematical sense?
6- page 10, line 238: change the typeface - the number of the reference article is too small.
7- page 11, figure 7: for easier reading, the authors should indicate A for the upper part of the figure and B for the lower part, also including it in the text.
8- page 11, line 260: did the authors check that the effect was indeed synergistic in the strict mathematical sense (the effect must be greater than the sum of the effects of the molecules taken separately)?
9-page 12, line 272: same comment on synergy
10-page 12, figure 8: can the authors explain the figures (-2/-2, -2....) in the legend?
11- page 16, table 2: I assume that LG is a ligand? Can the authors indicate this in the legend? The same applies to many of the figures in the manuscript
12- page 29, lines 679-687: paragraph not very clear, it would be advisable to reword it.
13- page 30, line 697: please note Fig. 27A instead of Fig. 27a.
14- page 30: part B of Fig. 27 is not written clearly. Can the authors reword it?
15- page 31, line 734: the authors indicate the presence of aggregation. Do the cells remain viable despite this aggregation? What is the quantity or % of PS that aggregates? Can tumour cells eliminate this aggregation?
16- page 38, line 910: did the authors check that the effect was indeed synergistic in the strict mathematical sense?
17- page 40, line 957: note Fig. 39 instead of figure 38
18- page 41, line 976: can the authors explain why "this dual selectivity minimized side effects in normal tissues"?
19- page 44, figure 43: the explanation of this figure should be reworded for greater clarity.
20- page 45, line 106: compounds 90 and 91 are not found in the figures. Could you please correct this?
Author Response
Dear Reviewer,
Thank you for your efforts in reviewing our manuscript and valuable feedback. Responding to the comments below, we revised the manuscript.
1) a table summarising all the acronyms would be welcome
The table with acronyms was added
2) Similarly, acronyms should be explained in all figure legends.
Since the table with acronyms has been provided, we believe that there is no need to repeat this information in figure legends.
3) page 9 line 190: there is an extra dash at the beginning of the line
Corrected
4) page 9 lines 211-226: this paragraph is not very clear, could the authors reword it?
Done
5) page 10, line 225: did the authors check that the effect was indeed synergistic in the strict mathematical sense?
We revised this sentence and replaced the synergistic effect with the bystander effect in accordance with the article being reviewed.
6) page 10, line 238: change the typeface - the number of the reference article is too small.
Corrected
7) page 11, figure 7: for easier reading, the authors should indicate A for the upper part of the figure and B for the lower part, also including it in the text.
Done
8) page 11, line 260: did the authors check that the effect was indeed synergistic in the strict mathematical sense (the effect must be greater than the sum of the effects of the molecules taken separately)?
The combination index was calculated by authors, and cytotoxic effect was greater than the sum of effects of individual molecules. The paragraph was updated with necessary information.
9) page 12, line 272: same comment on synergy
We revised this sentence and replaced the synergistic effect with the bystander effect in accordance with the article being reviewed.
10) page 12, figure 8: can the authors explain the figures (-2/-2, -2....) in the legend?
Done
11) page 16, table 2: I assume that LG is a ligand? Can the authors indicate this in the legend? The same applies to many of the figures in the manuscript
LG is a leaving group. We updated the legends.
12) page 29, lines 679-687: paragraph not very clear, it would be advisable to reword it.
The paragraph was revised.
13) page 30, line 697: please note Fig. 27A instead of Fig. 27a.
Corrected
14) page 30: part B of Fig. 27 is not written clearly. Can the authors reword it?
The paragraph was revised.
15) page 31, line 734: the authors indicate the presence of aggregation. Do the cells remain viable despite this aggregation? What is the quantity or % of PS that aggregates? Can tumour cells eliminate this aggregation?
The authors showed the presence of aggregation of dimeric and trimeric conjugates by measuring the absorption and fluorescence spectra. In addition, the authors did not determine the percentage of aggregation. They also did not show whether tumor cells can eliminate aggregation.
16) page 38, line 910: did the authors check that the effect was indeed synergistic in the strict mathematical sense?
As the required combination index was not calculated by the authors, the attributed effect could not be called synergistic. We revised the sentence.
17) page 40, line 957: note Fig. 39 instead of figure 38
Corrected
18) page 41, line 976: can the authors explain why "this dual selectivity minimized side effects in normal tissues"?
The authors did not evaluate the selectivity of the conjugate in comparison with normal cells. The sentence is revised.
19) page 44, figure 43: the explanation of this figure should be reworded for greater clarity.
Done
20) page 45, line 106: compounds 90 and 91 are not found in the figures. Could you please correct this?
Corrected